# Structures of active Hantaan virus polymerase uncover the mechanisms of *Hantaviridae* genome replication

Quentin Durieux Trouilleton [1,6], Sergio Barata-García[2,6], Benoît Arragain [1,5,6], Juan Reguera [2,3] ✉ & Hélène Malet [1,4] ✉

Hantaviruses are causing life-threatening zoonotic infections in humans. Their tripartite negative-stranded RNA genome is replicated by the multi-functional viral RNA-dependent RNA-polymerase. Here we describe the structure of the Hantaan virus polymerase core and establish conditions for in vitro replication activity. The apo structure adopts an inactive conformation that involves substantial folding rearrangement of polymerase motifs. Binding of the 5′ viral RNA promoter triggers Hantaan virus polymerase reorganization and activation. It induces the recruitment of the 3′ viral RNA towards the polymerase active site for prime-and-realign initiation. The elongation structure reveals the formation of a template/product duplex in the active site cavity concomitant with polymerase core widening and the opening of a 3′ viral RNA secondary binding site. Altogether, these elements reveal the molecular specificities of *Hantaviridae* polymerase structure and uncover the mechanisms underlying replication. They provide a solid framework for future development of antivirals against this group of emerging pathogens.

*B unyavirales* is a large viral order encompassing several zoonotic emerging viruses with high epidemic potential[1]. Notably, the *Hantaviridae* family, whose main hosts are insectivores and rodents, can infect humans and induce severe diseases[2]. Old world hantaviruses such as Hantaan (HTNV) and Puumala viruses are found in Asia and Europe where they cause hemorrhagic fever with renal syndrome that lead to death in up to 15% of the infections[3]. New world hantavirus such as Andes or Sin Nombre are prevalent in the Americas and are causing hantavirus cardiopulmonary syndrome with a mortality rate reaching 40%[4]. There are currently neither drugs nor vaccines to counteract these dangerous pathogens.

In this context, we have focused our interest on the replication of the tri-partite negative-stranded RNA genome of hantaviruses[5]. This essential step of the viral cycle is performed by a monomeric 246 kDa viral enzyme: the polymerase, also called L. As for other segmented

negative-stranded RNA viruses (sNSV), the replication of hantavirus genome is done de novo by the polymerase, in the absence of any primer. Initiation of replication on a viral RNA template (vRNA) has been suggested to occur internally at nucleotide 4 to produce a trinucleotide primer that is then realigned at the 3′ end of the genome[6]. This proposed mechanism, which needs to be validated structurally, would be made possible by a triplet nucleotide repetition at the 3′ end of the genome.

Structures of polymerases from several *Bunyavirales* families have been determined in the last few years including polymerases of La Crosse virus (LACV, *Peribunyaviridae*)[7,8], Severe Fever with Thrombocytopenia Syndrome virus (SFTSV, *Phenuiviridae*)[9,10], Rift Valley Fever virus (RVFV, *Phenuiviridae*)[11], Lassa virus (LASV, *Arenaviridae*) and Machupo virus (*Arenaviridae*)[12]. These structures share a common global organization with an N-terminal endonuclease domain (ENDO),

---

[1]Univ. Grenoble Alpes, CNRS, CEA, IBS, F-38000 Grenoble, France. [2]Aix-Marseille Université, CNRS, AFMB UMR, 7257 Marseille, France. [3]INSERM, AFMB UMR, 7257 Marseille, France. [4]Institut Universitaire de France (IUF), Paris, France. [5]Present address: European Molecular Biology Laboratory (EMBL), Grenoble, France. [6]These authors contributed equally: Quentin Durieux Trouilleton, Sergio Barata-García, Benoît Arragain.
✉e-mail: juan.reguera@inserm.fr; helene.malet@ibs.fr

followed by a central polymerase core containing the active site, and have a more variable C-terminal region that contains a Cap-binding domain (CBD) or a CBD-like domain. Some of these structures were determined in complex with the 5′ and 3′ vRNA promoters that are present at the genome ends. Nucleotides 1–10 of the 3′ and 5′ genome ends, despite being complementary, bind as single-stranded RNA in specific sites[7,12–14]. The 5′ binds as a hook in a specific site located on the surface of the polymerase core region, whereas the 3′ binds either in the active site or in a distinct 3′-end secondary binding site. Complementary nucleotides around positions 11 to 20 form a distal duplex that protrudes from the polymerase core. Structures stalled at specific stages of replication have then been determined for LACV-L[15], SFTSV-L[14], and LASV-L[13] providing key insight into the mechanisms underlying the genome replication of these viruses, notably revealing the mechanism of prime-and-realign initiation for LACV-L.

In this work, we present the near-atomic resolution structure of HTNV-L that gives insight into the organization of a polymerase from the *Hantaviridae* family, depicting molecular specificities compared to the structures of other *Bunyavirales* polymerases. We reveal that HTNV-L adopts an inactive conformation in the absence of RNA that notably highlights an unanticipated folding of the canonical polymerase motif E. We show that HTNV-L reorganization into a conformation compatible with activity is triggered by the binding of the 5′ vRNA, which itself leads to the recruitment of the 3′vRNA end towards the active site in a position suitable for prime-and-realign initiation. In vitro replication activity assays coupled with structures of HTNV-L stalled in defined active stages uncover the molecular basis of *Hantaviridae* replication.

## Results

### Structure determination of HTNV-L core

The N-terminal His-tagged full-length construct of HTNV-L was expressed in insect cells and purified to homogeneity (Supplementary Fig. 1). It was mutated in position 97 (HTNV-L$_{D97A}$) to abolish the ENDO activity[16] (Supplementary Table 1). To determine its structure, cryoelectron microscopy (cryo-EM) UltraAufoil grid preparation was optimized with PEGylation reagent to decrease preferential orientation and aggregation due to air-water interface interaction[17], resulting in grids suitable for single particle cryo-EM data collection (Supplementary Fig. 2a). Cryo-EM image analysis led to the determination an HTNV-L$_{D97A}$ apo structure at 3.3 Å resolution, revealing the organization of the polymerase core that is globally conserved compared to the one of other sNSV polymerases determined as far[7,9–12] (Fig. 1, Supplementary Figs. 2a, 3a, Supplementary Data 1, Supplementary Table 2). The flexibility of both the N-terminal ENDO domain and the C-terminal region precluded their structural characterization (Fig. 1). HTNV-L$_{D97A}$ core consists of a central region containing the canonical palm, finger and thumb domains of RNA-dependent RNA-polymerases (RdRp) that together form the central cavity containing the active site (Fig. 1b–d). The active site contains a loop in the palm domain (residue 923–927) that is analogous to the Prime-and-Realign loop (PR loop) shown in LACV-L to be important for replication initiation[15] (Fig. 1a, b). A different loop called "priming loop" (residues 1322–1334), based on the naming from the Influenza polymerase, is short and disordered in the apo conformation (Fig. 1a). As in other sNSV polymerases, several domains encircle the palm/finger/thumb central core (Fig. 1b, Supplementary Fig. 3). The finger and the palm are packed and stabilized by the linker region that connects the ENDO to the core. The ordered part of the linker region is simpler and shorter than its equivalent in other *Bunyavirales* polymerases, with a loop replacing a longer linker composed of several α-helices (Supplementary Fig. 4a). The thumb and the palm are buttressed by the core lobe that is analogous to Influenza PA-C. This region contains a potential vRNA-binding lobe (vRBL), which, by analogy with other sNSV polymerase structures[7,13,14,18], is susceptible to binding the 5′ and 3′ vRNA

promoters. The central core is also encircled by the bridge that interacts with the finger and the thumb-ring. Inserted in the thumb-ring is the lid that closes the active site cavity. The cavity is connected to the protein surface by 4 tunnels that correspond to the template entry, nucleotide entry, template exit, and product exit (Fig. 1d).

In the apo conformation, a positively charged pocket is present between the vRBL, the finger, and the core lobe in a conformation suitable to bind the 5′vRNA end (Fig. 1b, d, Supplementary Fig. 5a). A positively charged groove is also present at the polymerase surface between the vRBL and the thumb-ring, suggesting that it might correspond to the 3′vRNA secondary binding site (Fig. 1d, Supplementary Fig. 5a). In the apo conformation, the groove entrance is closed by two elements that we call the upper and the lower jaw (Fig. 1b, d). The upper jaw belongs to the vRBL domain and resembles the clamp found in LACV-L, although it is smaller[7]. The lower jaw belongs to the thumb-ring domain.

While global domain organization is conserved with other sNSV polymerases, two neighboring surface-exposed structural additions are specific to *Hantaviridae* polymerase: (i) a two β-strand insertion (residues 949 and 963) that corresponds to the α-helical Californian insertion observed in LACV-L and (ii) a two α-helix insertion between residues 1109 and 1137 (Supplementary Fig. 4b). The functional importance of these insertions remains to be determined. On the contrary, the α-ribbon that protrudes from the finger domain in the other *Bunyavirales* polymerase structures, which interacts with the distal duplex and undergoes significant movements during activity, is absent in HTNV-L (Supplementary Fig. 4c).

### 5′vRNA-binding activates HTNV-L through major conformational changes

To analyze promoter binding to HTNV-L, electrophoretic mobility shift assays (EMSA) were performed with fluorescently labeled RNA corresponding to the terminal 25 nucleotides of the 3′vRNA and 5′vRNA (3′ vRNA$_{1–25}$ and 5′vRNA$_{1–25}$) (Fig. 2a). HTNV-L was incubated with an unspecific poly-A RNA prior to addition of fluorescent 5′vRNA$_{1–25}$ and 3′vRNA$_{1–25}$. Competition between non-specific poly-A RNA and fluorescent vRNA ensures visualizing specific vRNA interactions to HTNV-L. In these conditions, interaction with 5′vRNA$_{1–25}$ was clearly detected, whereas 3′vRNA$_{1–25}$ binding was very poor, consistently with the presence of (i) a pre-formed 5′ binding site and (ii) a closed 3′ secondary binding site in the HTNV-L apo structure (Fig. 1b, d). 5′vRNA$_{1–25}$ binding through its single-stranded 5′-end was suggested by EMSA assays of the 5′vRNA$_{1–25}$ incubated with complementary DNA of various lengths (Fig. 2b). It identified 5′vRNA$_{1–25}$ binding only when a minimum of 10 nucleotides were left single-stranded on the 5′-end. The 5′vRNA$_{1–25}$-bound HTNV-L$_{D97A}$ cryo-EM structure was determined at 3.2 Å resolution and reveals the binding of the 5′vRNA as a hook formed by nucleotides 2 to 12 (Fig. 2c, d, Supplementary Figs. 2b, 6a, Supplementary Table 2). The 5′vRNA nucleotide 1 is visible at the protein surface but is quite flexible, nucleotides 13–25 are flexible and not visible. The hook is stabilized by the canonical base pairing G3/C11 and U4/A10, while A2 and A12 interact through H-bonds. U12 replaces A12 in some HTNV strains, strongly suggesting a third canonical base pairing in these cases. The 5′vRNA hook binds to a pocket formed by the vRBL, the core lobe, the finger, and the fingernode through specific stacking, polar, and H-bond interactions (Fig. 2d). The RNA binding site is covered by an arch (residues 391–398) that protrudes from the vRBL but remains partially disordered (Fig. 2c).

Comparative analysis of HTNV-L apo and HTNV-L 5′vRNA$_{1–25}$-bound map reveals unexpected conformational changes with a switch from a previously unknown apo structure incompatible with activity to a 5′ vRNA$_{1–25}$-bound map with motif organization compatible with activity (Fig. 2e). A global comparison reveals that the motif E adopts an unusual α-helical configuration in the apo map and switches to the canonical β-hairpin strand structure upon 5′vRNA binding

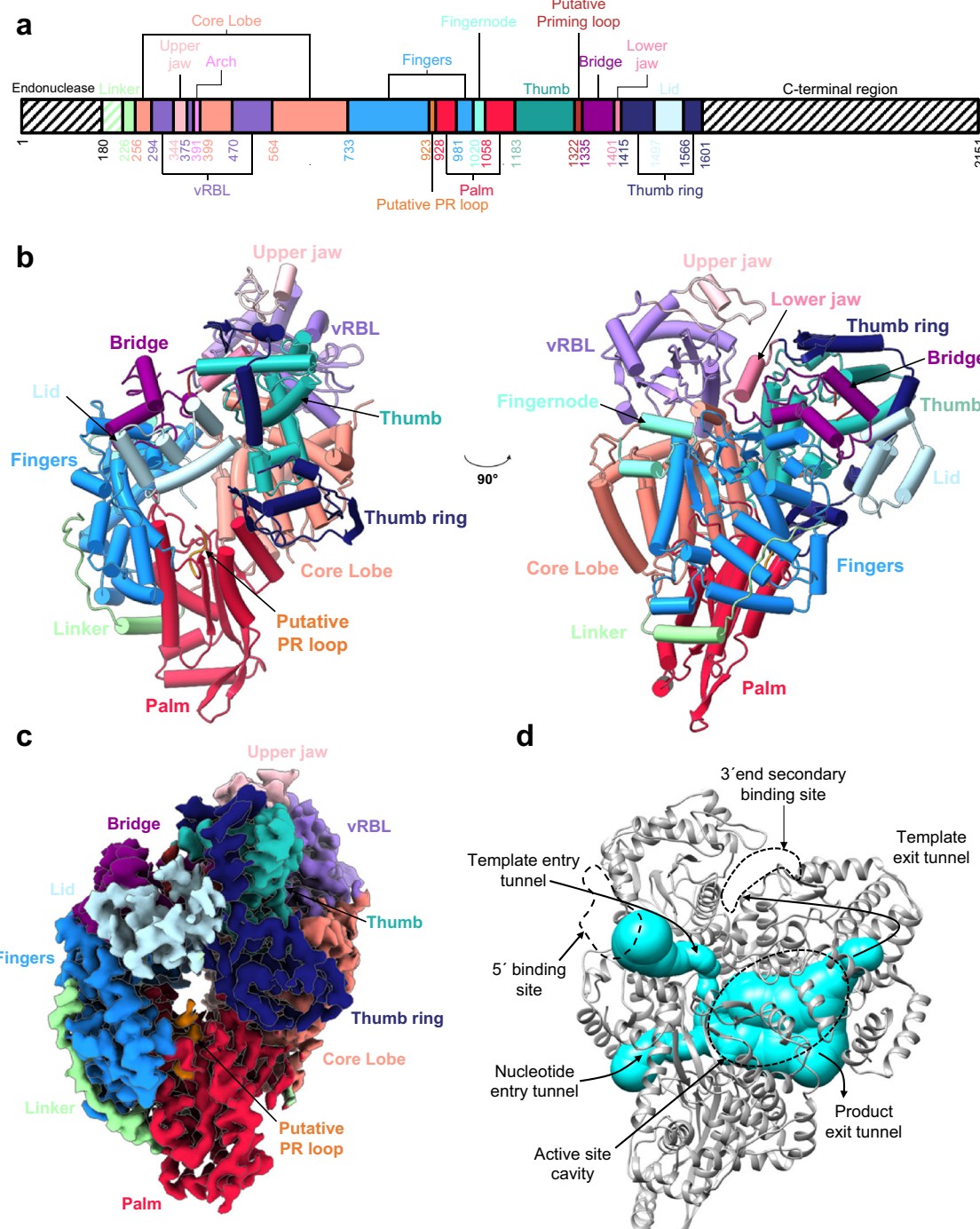

**Fig. 1 | Structural organization of HTNV-L. a** Schematic representation of HTNV-L domain structure. **b** Two orthogonal views of apo HTNV-L model shown as cartoon. The domains are indicated with arrows and colored as in **a. c** Cryo-EM map of apo HTNV-L colored as in **b. d** HTNV-L in the same orientation as in the right panel of **b** showing the four tunnels that reach the active site cavity as cyan volumes. The binding sites of the 5′vRNA and the 3′vRNA secondary site are indicated.

(Supplementary Fig. 7). Motif E is positioned between the palm and the thumb where it acts as a hinge, with its reorganization inducing a global movement of the palm domain (Supplementary Movie 1). A careful analysis reveals the cascade of reorganization occurring between the apo and the 5′vRNA$_{1-25}$-bound maps (Fig. 2f, g, Supplementary Fig. 8a). 5′vRNA$_{1-25}$ interaction is inducing a movement of the fingernode, the core lobe finger linker and the first helix of the finger (733–758). This provides space and charges resulting in the ordering of fingertip residues 886–890, which were disordered in the apo map. Fingertip organization upon 5′vRNA$_{1-25}$ binding induces the

stabilization of motif B (Fig. 2f, Supplementary Fig. 8a). This triggers the reorganization of motifs A, C, E and of the PR loop that are all in proximity and interconnected. These motifs, which were all misplaced in the apo map, position themselves upon 5′vRNA binding in conformations compatible with replication activity (Fig. 2f, g, Supplementary Fig. 8a, b). The movement is particularly drastic for the residue E1170 of motif E that changes its position by 12.7 Å between the apo and the 5′vRNA$_{1-25}$-bound configuration. As a result of its repositioning, E1170 can participate, together with D972 (motif A) and D1099 (motif C), to the binding of a magnesium ion in the active site in 5′

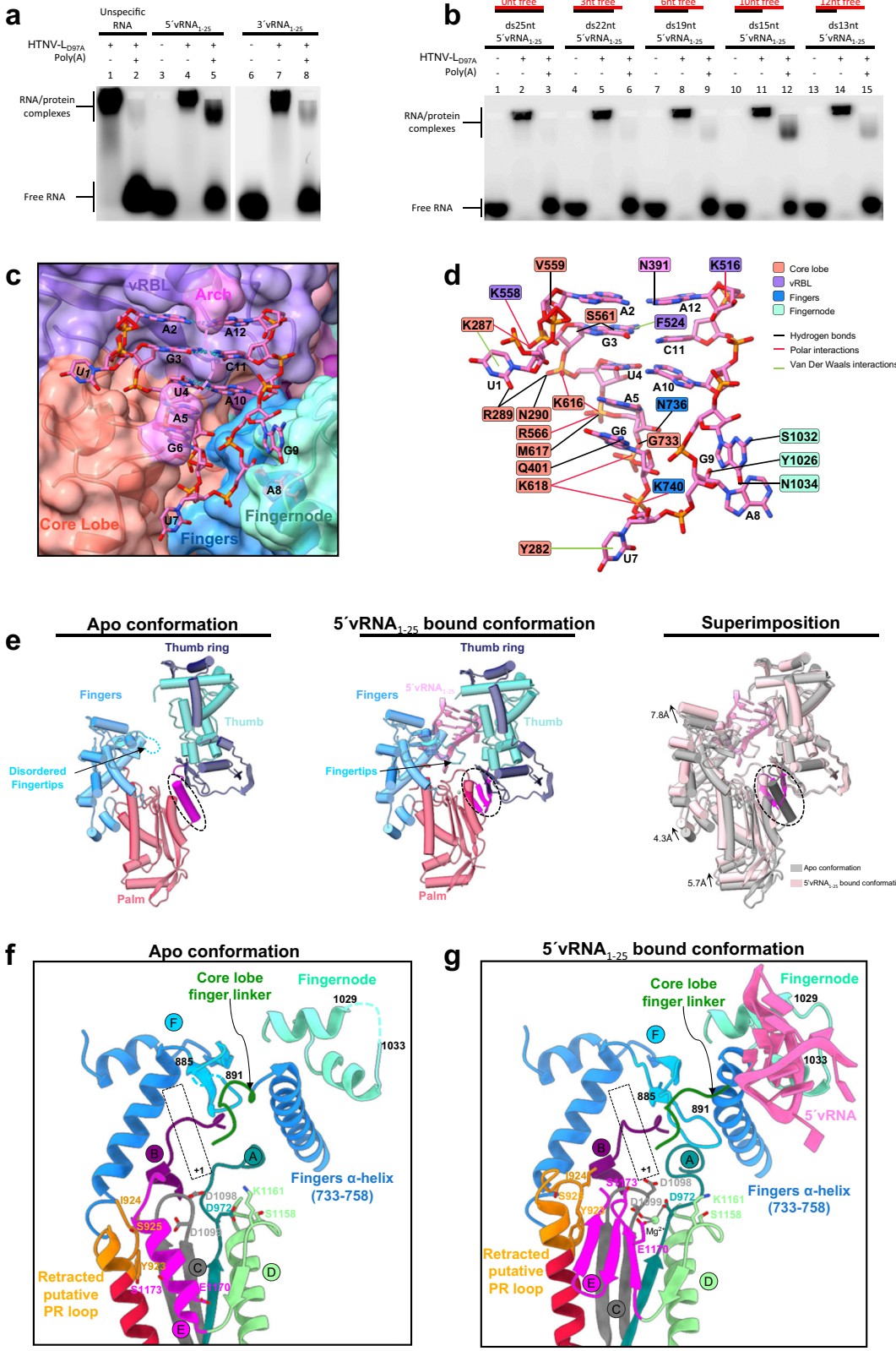

vRNA$_{1-25}$-bound conformation (Fig. 2g, Supplementary Fig. 8b). The PR loop tip also undergoes a large displacement of 6 Å between the apo and the 5'vRNA$_{1-25}$-bound map, it orders itself to reach its canonical retracted position observed in most active sNSV polymerase structures (Fig. 2g, Supplementary Fig. 8b, c). Altogether, these local and coordinated modifications result in a global reconfiguration of HTNV-L to form of a proper active site compatible with replication activity.

## HTNV-L actively replicates 3'vRNA template in vitro

Structure activation by the 5'vRNA$_{1-25}$ binding prompted us to analyze if the 5'-end would recruit the 3' template towards the active site for replication initiation. To avoid the formation of double-stranded RNA that would prevent the binding of the 5'vRNA nucleotides 1–12 as a hook in HTNV-L$_{D97A}$ (Fig. 2b), the 5'vRNA was carefully mutated and designed to reduce 3'/5' complementary while maintaining

**Fig. 2 | 5′vRNA binding triggers HTNV-L reorganization and activation.**
**a** Electrophoretic mobility shift assays showing 5′ and 3′vRNA binding to HTNV-L. Migration position of free RNA and RNA-bound to HTNV-L are labeled. Conditions where poly(A) RNA was added to limit RNA unspecific interaction are indicated. 5′ vRNA and the unspecific RNA are labeled with FAM fluorophore. 3′vRNA is labeled with Cy5 fluorophore. Source data is provided as a Source Data file. **b** EMSA assay with the 5′vRNA$_{1-25}$ incubated with complementary DNA of various lengths. A schematic above indicates the number of bases left single-stranded on the 5′end of the 5′vRNA$_{1-25}$. Conditions with poly(A) are indicated as in **a**. In the schematics, RNA is colored in red and DNA in black. Source data is provided as a Source Data file. **c** 5′ vRNA hook binding site. The nucleotides 1 to 12 of the 5′vRNA$_{1-25}$ are represented in pink and the hydrogen bonds linking the bases of the hook are indicated. HTNV-L is shown as a transparent surface to visualize the binding cavity. Regions involved in

the binding site formation are labeled and shown as cartoon. The part of the arch that is disordered is indicated with a dotted line. **d** HTNV-L residues that bind to the 5′vRNA$_{1-25}$ hook are labeled. Hydrogen bond, polar and van der Waals interactions are respectively shown as black, red, and green lines. **e** The central domains of HTNV-L are colored as in Fig. 1 and displayed as cartoon for both the apo and the 5′ vRNA$_{1-25}$-bound conformations. The motif E is colored in magenta and surrounded by a dotted line. A superimposition of the apo and the 5′vRNA$_{1-25}$-bound conformations is shown. Arrows indicate movements and distances between the apo and the 5′vRNA$_{1-25}$-bound states. **f, g** Zoom on the motifs, the putative prime-and-realign (PR) loop, the 5′vRNA-binding site in the apo conformation (**f**) and the 5′ vRNA$_{1-25}$-bound conformation (**g**). Each element is labeled and has a specific color. The position of the active site that corresponds to the first nucleotide to be incorporated is labeled as +1.

5′/HTNV-L binding. The base pairing U4/A10 was modified into G4/C10 as the bases 4 and 10 do not interact directly with HTNV-L (Fig. 3a). Since careful analysis of 5′vRNA composition identified the base of nucleotide 9 as a uridine in some HTNV strains, we mutated it accordingly to decrease the 5′/3′ complementarity (Fig. 3a). EMSA assays clearly show that 5′mut binds to HTNV-L as well as 5′ WT (Fig. 3a).

To analyze if 5′mut was compatible with in vitro replication of HTNV-L$_{D97A}$, activity assays were performed. HTNV-L$_{D97A}$ was incubated with the 5′mut, 3′vRNA$_{1-25}$, NTPs and MgCl$_2$ for 4 h at 30 °C. Faint products were observed with the main product being a 24-mer nucleotide product (Fig. 3b, lanes 1 and 2, [$^{32}$P-GTP] was multiplied by 3 in lane 2 compared to line 1 to boost the signal and clearly visualize the product). As a control, a reaction with 5′WT instead of 5′mut (Fig. 3b, lane 3) did not generate any replication product, further validating that mutation of the 5′ hook is necessary to observe replication in vitro. We then assessed if addition of dinucleotide primers would generate equivalent products and increase product formation as previously reported for sNSV polymerases[10,18–20]. We therefore added UpA, ApG, or GpU to the replication reactions (Fig. 3b, lanes 4 to 6). These dinucleotides are respectively complementary to nucleotides 1–2, 2–3, and 3–4 of the template 3′-end, and can also respectively bind to nucleotides 4–5, 5–6, 6–7 due to triplet repetition at the 3′-end of the template (Fig. 3d, e). The three primers increased the formation of products, with ApG being the most efficient primer with the formation of 24-mer and 21-mer replication products (Fig. 3b, lane 5). 21-mer product can be interpreted as a hybridization of ApG with nucleotides 5–6 of the template and an elongation without realignment (Fig. 3d). 24-mer products can be due either to (i) elongation without realignment of a primer that would hybridize with nucleotides 2–3 of the template or (ii) hybridization with 5–6 of the template followed by realignment as previously proposed[6] (Fig. 3e). A one nucleotide difference in the products generated with UpA, ApG, and GpU is visualized, and is consistent with the extension of primer that will dictate the product length (Fig. 3d, e). UpA is much less efficient than ApG as a primer and produces 25- and 22-mer main products (Fig. 3b, lane 4). GpU is almost as efficient as ApG as a primer and produces main 23-mer products (Fig. 3b, lane 6). We also visualize faint product elongated by ~3 nucleotides (~27-mer above the 24-mer main product on line 5, ~26-mer above the 23-mer product on line 6). These faint products could correspond to a double prime-and-realign reaction that would result in an extra-triplet incorporation at the 5′-end of the product.

We also tested if the replication could be stalled at early-elongation by incubating HTNV-L with a 3-nucleotide subset (Fig. 3b, lane 7). This is expected to result in the generation of a 9-nucleotide product that corresponds to nucleotides 2 to 10 of the template, as (i) ApG primer corresponds to position 2 of the template and (ii) the absence of CTP in the mix induces a stop at position 11 of the template that is a G. This nicely confirms the presence of the expected 9-mer product. We also visualize ~6-mer and ~11-/12-mer products that could

correspond to un-realigned and two-times realigned products, respectively, generated by imperfect prime-and-realign initiation in vitro. CTP addition after 1 h restores complete product formation (Fig. 3b, lane 8) indicating that HTNV-L$_{D97A}$ was stalled in an active early-elongation conformation.

The ability to visualize a faint signal in the absence of any primer prompted us to analyze whether the products were mono-phosphate, as previously proposed[6]. We therefore incubated the product with a 5′terminal terminator exoribonuclease (ExoU) that specifically cuts monophosphate 5′RNA but not tri-phosphate RNA (Fig. 3c). As a control, the 25-nt molecular weight marker that contains a 5′mono-phosphate due to its labeling with T4 nucleotide kinase, was incubated with ExoU. The positive control was cleaved as expected, but HTNV-L$_{D97A}$ replication product generated in the absence of dinucleotide marker was not. We can therefore conclude that, in the conditions used in the in vitro assays, the product formed has a tri-phosphorylated 5′ end.

## Pre-initiation state: 3′vRNA in position for prime-and-realign initiation

To reveal how the 5′vRNA recruits the 3′ template towards the active site for its replication, we incubated HTNV-L$_{D97A}$ with 5′mut and 3′ vRNA$_{1-25}$ at 30 °C for 1 h and subsequently determined its cryo-EM structure at 3.3 Å resolution (Fig. 4a, b, Supplementary Fig. 6b and 9, Supplementary Table 2). In this structure, the 5′mut nucleotides 1–12 bind as the WT 5′vRNA, thereby validating the mutation strategy (Supplementary Fig. 5b, c). 5′mut binding plays an important role in the proper positioning of the 3′ vRNA end in the active site by anchoring the 5′ in a specific position of HTNV-L$_{D97A}$. The next nucleotides of the 5′, 13 to 20, form a distal duplex with their equivalent, almost complementary, nucleotides of the 3′vRNA. The distal duplex interacts with residues 361–366 of the upper jaw and 372–391, 492–496 of the vRBL, bringing the nucleotides 12 to 1 of the 3′vRNA$_{1-25}$ towards the active site (Fig. 4a, b). The nucleotides 12 to 10 position themselves in the template entry tunnel and interact with residues 492 and 519–521 from the vRBL, 1341–1344, and 1396–1403 from the bridge (Fig. 4a, c). The next nucleotides are positioned in a large buffer zone of the entry tunnel that allows for flexibility and therefore prevents visibility of a few nucleotides (Fig. 4a–c). Subsequent nucleotides are visible at the end of the entry tunnel, in the active site (in position −1 and position +1) and overstep the active site up to position −4, positioning themselves close to the putative PR loop (Fig. 4d). Although visible, the nucleotides from the entry tunnel (position +3) to the position −4 of the active site are less defined. Their phosphate and ribose have defined densities whereas their bases cannot be unambiguously identified, suggesting that the nucleotides may have a register that varies between the particles as this is permitted by the flexibility of nucleotides in the buffer zone. This RNA configuration is compatible with a prime-and-realign mechanism in which the replication would initiate internally, generating a primer that could be then realigned thanks to the buffer zone.

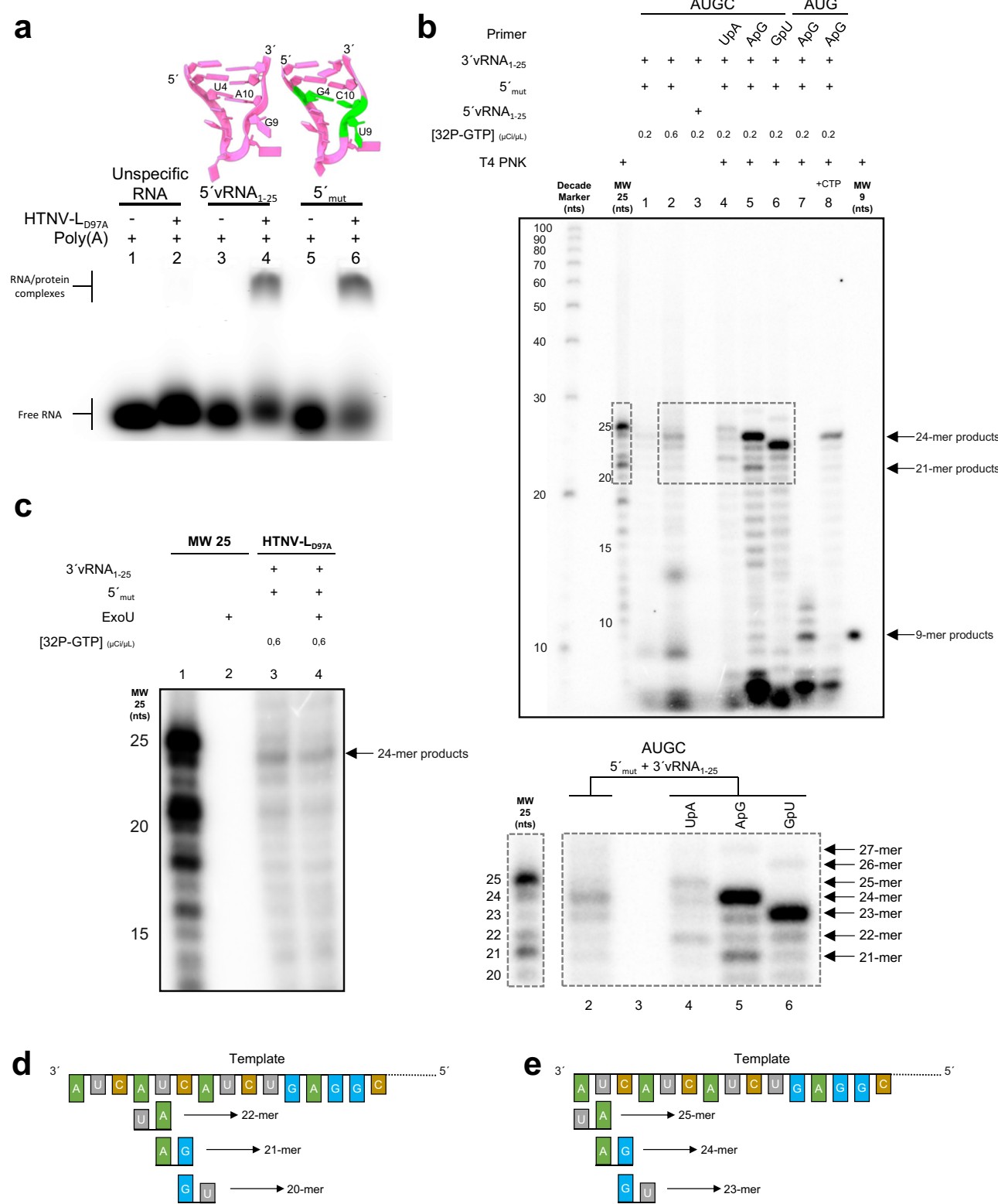

Comparison of the 5′vRNA$_{1–25}$-bound and the pre-initiation maps reveals concerted movements related to 3′vRNA binding (Supplementary Fig. 10). Residues 1341–1344 of the bridge (shown in purple) and residues 1399–1402 of the bridge-lower jaw linker (shown in gray) become ordered in pre-initiation and interact with the 3′vRNA nucleotides 10 and 11 at the template entry tunnel. This triggers a movement of the lower jaw and the thumb-ring β-hairpin (residues 1414–1425, shown in dark blue) associated with small movements of

the thumb and thumb-ring. Opening of the bridge and the lid is also observed. As a result, the bridge loop (residues 1335 to 1341, in green) that follows the priming loop, becomes ordered and locates itself close to the lower jaw, the thumb-ring β-hairpin and the thumb. The priming loop itself remains partially disordered but is clearly located on the internal side of the active site cavity. In this configuration the active site cavity is too small to accommodate a template/product duplex and the template exit tunnel is closed.

**Fig. 3 | HTNV-L replication activity. a** Electrophoretic mobility shift assay showing the binding of 5′vRNA$_{1-25}$ and mutated 5′RNA (5′mut) to HTNV-L. On top, structure of the 5′vRNA$_{1-25}$ and 5′mut with mutated nucleotides shown in green. Source data is provided as a Source Data file for **a–c. b** In vitro replication assays. "+" signs indicate the presence of 3′vRNA$_{1-25}$, 5′mut, 5′vRNA$_{1-25}$ in the reaction. When dinucleotide primers were used, their nature is indicated. When indicated, T4 polynucleotide kinase (T4 PNK) was used at the end of the reaction to 5′mono-phosphorylate the products, ensuring their migration as the molecular weight markers. [$^{32}$P-GTP] is 0.2 µCi/µl for all the replication reactions except for lane 2 where it is 0.6 µCi/µl to boost the signal. MW25 corresponds to the 25-mer nucleotides (nts) RNA product that is complementary to the 3′vRNA$_{1-25}$. MW9 corresponds to the region complementary to nucleotides 2 to 10 of the 3′vRNA$_{1-25}$. RNA lengths of the MW25 are indicated on the left. Lengths of main products are indicated with arrows on the right of the gel. Dotted lines indicate the regions that were zoomed at the bottom of the panel. RNA lengths of the MW25 are indicated on the left of the zoomed gel. Lengths of products are indicated with arrows on the right of the zoomed gel. **c** Analysis of the nature of the 5′ end. Lane 1: 25-mer molecular weight marker that has a radiolabeled 5′ monophosphate (MW25). Lane 2: migration of MW25 incubated with the 5′ Terminator Exonuclease (ExoU) that specifically cuts 5′ mono-phosphate RNA (ExoU positive control). Lanes 3–4: HTNV-L replication assays with 5′mut and 3′vRNA$_{1-25}$ in the absence of dinucleotide primer. In lane 4, an incubation with the ExoU is performed. **d, e** Schematic indicating the possible binding sites of the dinucleotide primers on the 3′vRNA$_{1-25}$ template. Direction of replication is indicated with an arrow. Theoretical replication product lengths using a 25-mer template are indicated. They depend on the primer hybridization position. **d** corresponds to internal positioning of the primers, followed by elongation without realignment. **e** corresponds either to internal initiation with realignment or to terminal initiation.

## HTNV-L at elongation reveals template-product duplex formation

To reveal the necessary HTNV-L reorganization at elongation, we incubated HTNV-L$_{D97A}$, 5′mut and 3′vRNA$_{1-25}$ in a ratio 1:10:10, together with ApG primer, ATP, UTP, and GTP for 4 h at 30 °C. Cryo-EM data collection and image processing resulted in the determination of the elongation complex structure at 3.1 Å resolution (Supplementary Fig. 6c–e,11, Supplementary Table 2). The switch from initiation to elongation results in the disruption of the distal duplex due to the movement of the template towards the active site, progressively forming the product (Fig. 5a, b). Stalling of the product is observed after the incorporation of nucleotide 10 due to the presence of a G in template position 11 and the absence of CTP in the mix (Fig. 5a, b). As a result, the product is in a post-incorporation post-translocation state with the product active site position +1 left empty (Fig. 5c). The template/product duplex contacts many residues lining the active site chamber via both van der Waals and polar interactions (Fig. 5d, e). Y1564 of the lid positions itself in front of the most distal nucleotide of the product, preventing further growth of the duplex template product, thereby separating both strands for their subsequent exit as single-stranded RNA through their respective charged exit tunnels.

Template-product duplex formation results in the widening of the HTNV-L$_{D97A}$ core (Fig. 6a). The finger and the bridge slightly rotate to avoid clashes with the product. The thumb, the thumb-ring and the lid undergo larger rotation. Reorganization and rotation of the lower jaw leads to a movement of the thumb-ring β-hairpin and the bridge loop. These concerted rearrangements provide space for the priming loop to extrude from the active site cavity and positions itself between the lid and the thumb-ring. Altogether these movements result in the opening of the template exit channel (Fig. 6b).

## The 3′vRNA secondary binding site opens at elongation

Movements between pre-initiation and elongation states also result in (i) the formation of a charged path from template exit channel towards the 3′vRNA secondary binding site (Supplementary Fig. 12) and (ii) the opening of the 3′vRNA secondary binding site (Fig. 6c). These conformational changes are compatible with the binding of the end of the 3′vRNA template in the 3′vRNA secondary binding site at late elongation, following its exit through the template exit tunnel.

Opening of the 3′vRNA secondary binding site entrance is governed by the lower jaw helix (Fig. 6c). Comprising residues 1403 to 1411 at pre-initiation, the lower jaw helix extends to encompass residues 1401 to 1411 at elongation and its orientation is modified (Fig. 6c). As a result, residues that were closing the 3′RNA secondary site at initiation move by 5–6 Å, providing space for the 3′vRNA to enter. A movement of the upper jaw is also observed at elongation, with its tip residues acting as a clamp to further form the 3′vRNA secondary binding site cleft. As a result, seven terminal nucleotides of the 3′vRNA end can bind in an extended single-stranded configuration in a narrow cleft formed by the thumb, the core lobe, the vRBL, the lower and the upper jaw (Fig. 6c–e). Although there is to an excess of 3′vRNA$_{1-25}$ in the replication elongation reaction, they mimic 3′vRNA end binding at late elongation. Their interaction involves several specific binding pockets, burying a surface of 2590 Å$^2$ and making 12 protein-RNA H-bonds (Fig. 6d,e, Supplementary Table 3). Sequestration of the 3′ end in the 3′ vRNA secondary binding site at late elongation is likely to ensure efficient recycling of the 3′vRNA template for future rounds of replication.

## Discussion

Structures of HTNV-L determined in key, carefully chosen, states uncover several aspects that are interesting to compare with other known viral polymerase structures.

Of particular significance is the conformation of HTNV-L in its apo form, that had, to our knowledge, never been observed for any viral RdRp. The observed folding of motif E as an α-helix has a significant impact in the relative position of the thumb and the palm, which induces a differential placement of canonical motifs and inactivity of the polymerase (Fig. 7). Hinge movements between the palm and the thumb have been observed and were shown to be important for the activity of several viral polymerases, such as HIV reverse transcriptase[21] or Flavivirus polymerase[22,23]. Inhibitors targeting a pocket that is proximal to motif E has been described for Dengue and Zika polymerases[24,25]. The identified apo conformation, therefore, opens ways towards future drug developments that would target pockets located close to motif E and would block the polymerase in its inactive state.

We observe that binding of the 5′vRNA$_{1-25}$ triggers concerted local movements that lead to a global reconfiguration into a form compatible with activity (Fig. 2e–g, Fig. 7). The importance of 5′vRNA binding in stabilizing the fingertips and the motif B had already been seen for LACV-L, but in the latter other motifs were already in the pre-formed active conformation[7].

Activation of HTNV-L by 5′mut followed by incubation with the 3′vRNA$_{1-25}$ triggers the positioning of the 3′vRNA$_{1-25}$ in the active site. The only structures of sNSV polymerases with an RNA reaching the active site determined so far are the replication initiation structure of LACV-L[15] and the pre-initiation structure of bat influenza A[26] (Supplementary Fig. 13b,c). In LACV-L, the 3′vRNA end is stabilized in position −3 of the active site by the putative PR loop that extends at initiation. In Influenza, the priming loop is stabilizing the 3′vRNA end[26]. In the HTNV-L pre-initiation structure visualized here, the 3′vRNA$_{1-25}$ overshoots the active site, with the 3′vRNA$_{1-25}$ end located in position −4 of the active site, in accordance with a prime-and-realign mechanism of replication initiation from a vRNA template (Fig. 7). The priming loop remains partially disordered far from the active site and is unlikely to be involved in the stabilization of the 3′vRNA end due to its position and its relatively small size (Supplementary Fig. 13a). HTNV-L$_{D97A}$ putative PR loop remains in its resting retracted conformation at pre-

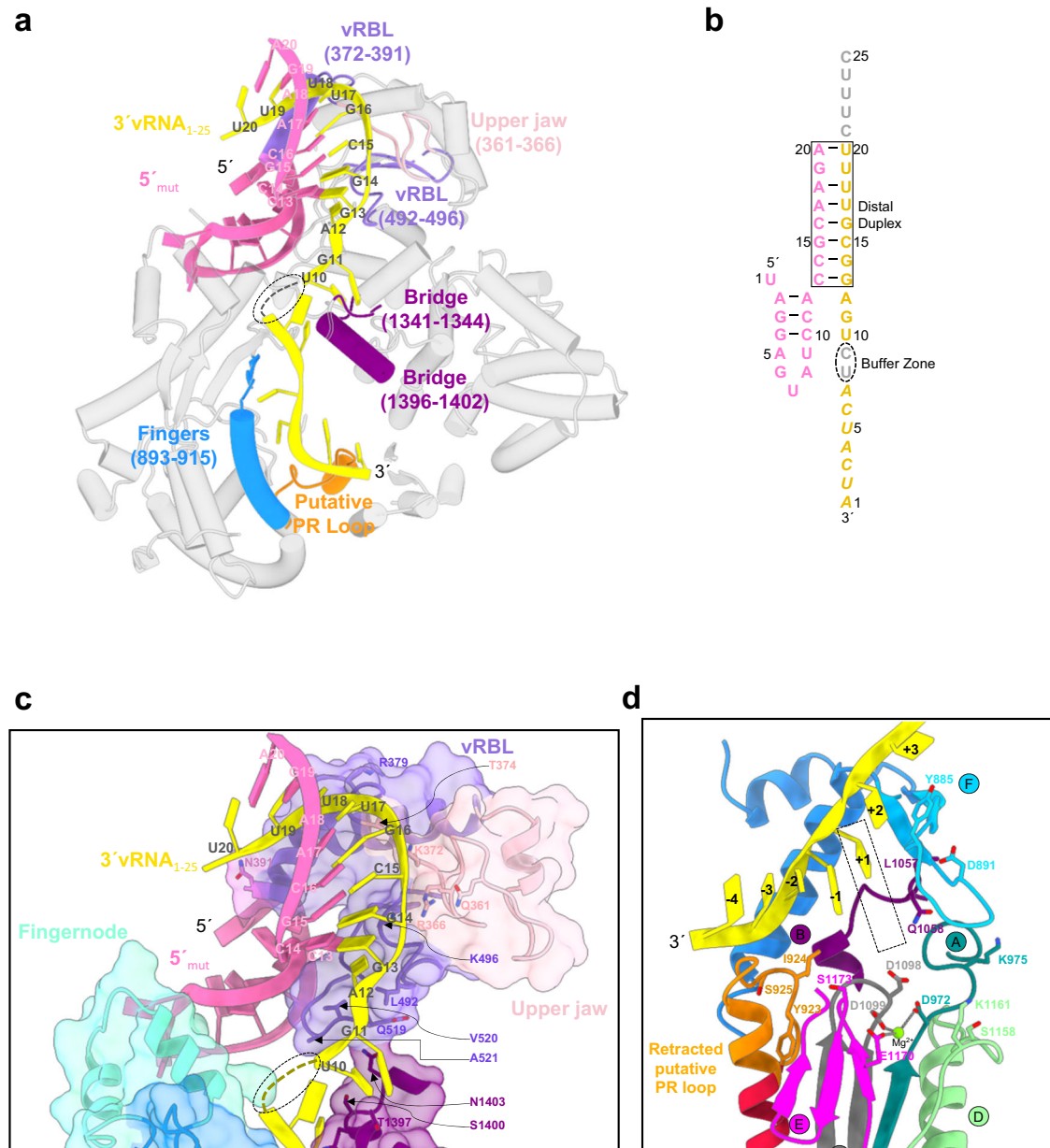

**Fig. 4 | HTNV-L in pre-initiation state. a** Cut-away global view of HTNV-L in pre-initiation state. The mutated 5′RNA (5′mut) and 3′vRNA$_{1-25}$ are respectively colored in pink and yellow. HTNV-L is shown as a semi-transparent white cartoon except in regions that directly interact with the RNA that are shown as non-transparent and are labeled. The buffer zone of the 3′vRNA$_{1-25}$ is indicated with a dotted line. **b** Schematic organization of the 5′mut and the 3′vRNA$_{1-25}$. RNA nucleotides that are present but not visible are shown in gray. Nucleotides that are visible but for which the register is not clear in the structure are shown in italics. The canonical base pairing between nucleotides is indicated with a line. The distal duplex and the buffer zone are surrounded. **c** Zoom on the interaction of the distal duplex and the buffer zone with HTNV-L. Surface interaction are displayed and colored as in Fig. 1. Residues that directly interact with the 3′vRNA$_{1-25}$ are indicated and labeled. Nucleotides of the buffer zone that display less interaction with HTNV-L and are disordered are surrounded by a dotted line. **d** Zoom on the active site at pre-initiation. The motifs and the putative PR loop are shown and colored as in Fig. 2f. The 3′vRNA$_{1-25}$ is shown as a yellow cartoon, it overshoots the active site to position itself close to the prime-and-realign (PR) loop visible in its retracted position. The position of the first nucleotide to be incorporated is labeled as +1. Important residues of the motifs and the putative PR loop are shown as stick and labeled. The magnesium ion is shown as a green dot.

initiation and it remains to be investigated whether it extends at initiation (attempts to structurally capture HTNV-L$_{D97A}$ initiation remained unsuccessful). According to HTNV-L replication pre-initiation map, the ApG primer used in the replication reaction would bind positions −2 and −3 of the active site, similarly to what was observed in LACV-L, suggesting a similar strategy of replication initiation.

Interestingly, replication assays performed in the absence of primer generated 5′ tri-phosphorylated products one nucleotide shorter than the template (Fig. 3c). RNA-sequencing of the band corresponding to the unprimed product is not realistic as the product is very faint and migrates at the same height as the template, precluding efficient signal detection. Several elements however lead us to hypothesize on the nature of the product. Indeed, the UpA-primed replication (Fig. 3b,

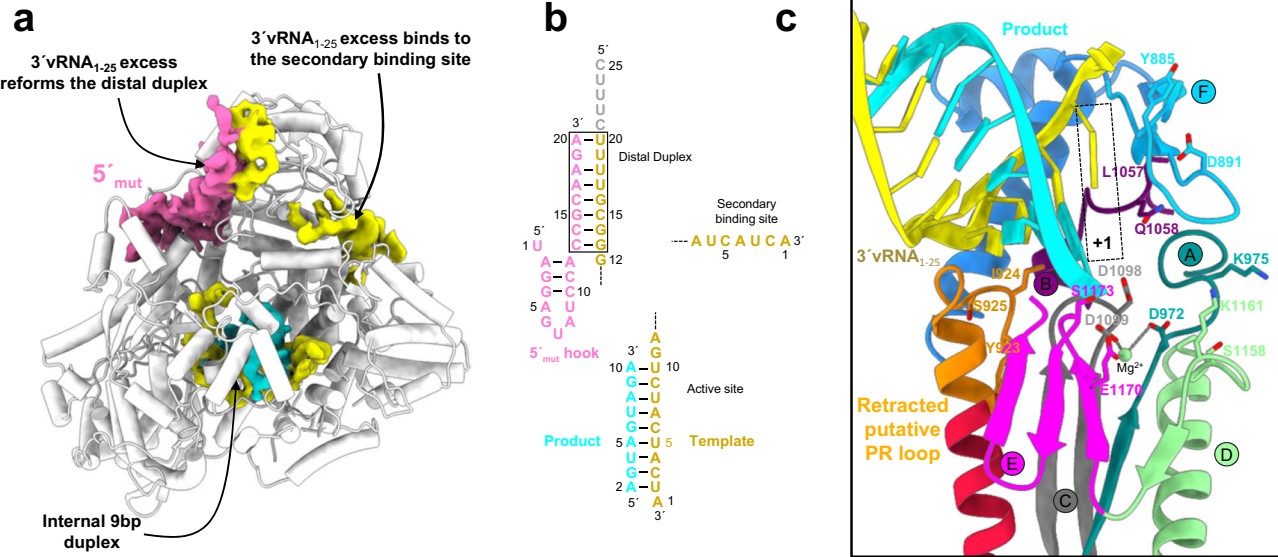

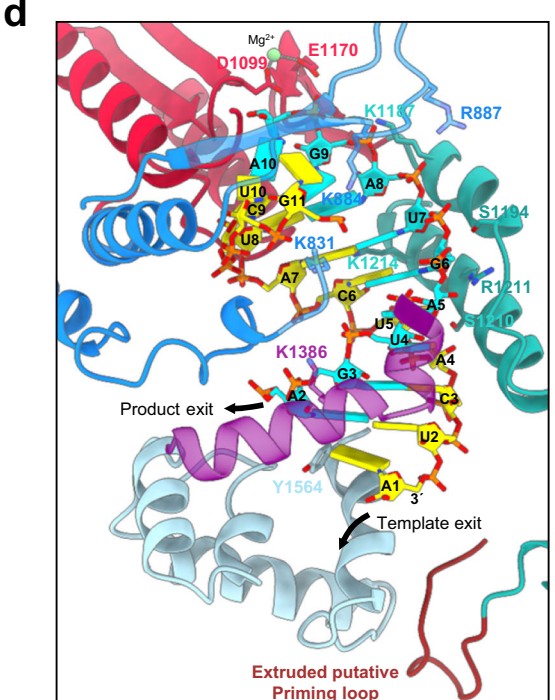

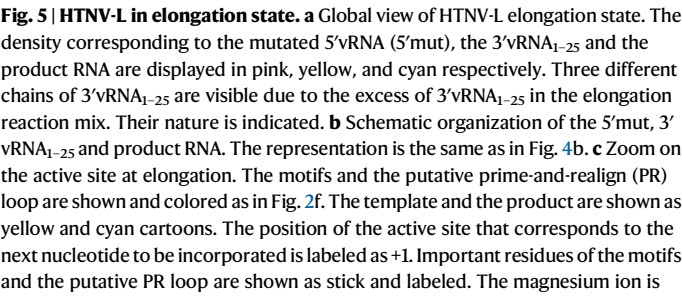

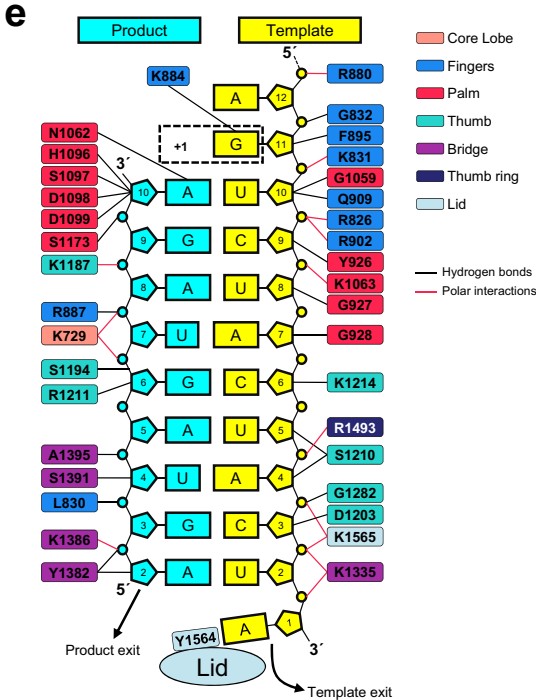

**Fig. 5 | HTNV-L in elongation state. a** Global view of HTNV-L elongation state. The density corresponding to the mutated 5'vRNA (5'mut), the 3'vRNA$_{1-25}$ and the product RNA are displayed in pink, yellow, and cyan respectively. Three different chains of 3'vRNA$_{1-25}$ are visible due to the excess of 3'vRNA$_{1-25}$ in the elongation reaction mix. Their nature is indicated. **b** Schematic organization of the 5'mut, 3' vRNA$_{1-25}$ and product RNA. The representation is the same as in Fig. 4b. **c** Zoom on the active site at elongation. The motifs and the putative prime-and-realign (PR) loop are shown and colored as in Fig. 2f. The template and the product are shown as yellow and cyan cartoons. The position of the active site that corresponds to the next nucleotide to be incorporated is labeled as +1. Important residues of the motifs and the putative PR loop are shown as stick and labeled. The magnesium ion is shown as a green dot. **d** Zoom on the template/product duplex in the active site cavity. The template and the product are shown as yellow and cyan cartoons with nucleotides labeled. HTNV-L regions that interact are displayed as cartoon and colored as in Fig. 1. Residues in interaction are shown as sticks and labeled. The template and product exits are shown with arrows. **e** Schematic representation of the HTNV-L/template/product interactions. HTNV-L residues that bind to the template and the products labeled are colored according to the domain to which they belong. Hydrogen bond and polar interactions are respectively shown as black and red lines. Position +1 of the active site is indicated. The lid domain and in particular its residue Y1564 that separates the template/product duplex is shown.

lane 4) is much less efficient than ApG-primed replication (Fig. 3b, lane 5), and unprimed products (Fig. 3b, lane 2) resemble ApG-primed products (Fig. 3b, lane 5). Our hypothesis is that unprimed products start at nucleotide 2, thereby generating pppApGpU… products. As

replication in vivo produces full-length products that are 5' mono-phosphorylated[6], one can hypothesize that monophosphorylated U could be added at the 5' end of the product at a later stage. Confirming or contradicting this idea will be the subject of future research. It will

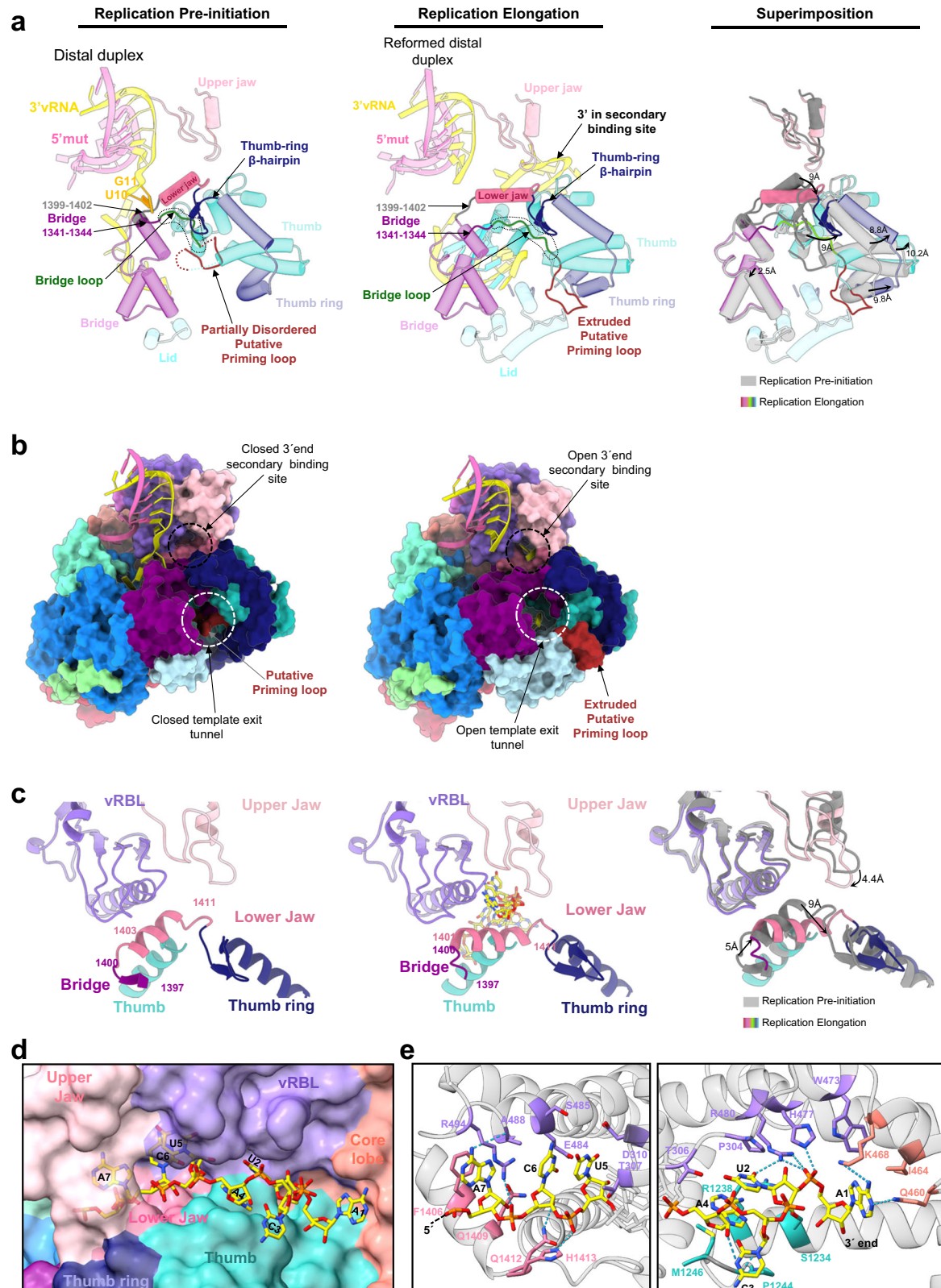

also be interesting in the future to investigate the mechanisms and the protein(s) involved in this process, which could be HTNV-L itself, another HTNV viral protein or a cellular partner.

While elements remain to be investigated regarding the 5'end of the replicated product, the HTNV-L$_{D97A}$ replicating structures unveil the mode of binding of the 3'vRNA template following its passage through the active site. Indeed, switching from initiation to elongation

leads to core widening and opening of the 3'RNA secondary binding site (Fig. 7, Supplementary Fig. 14a). This differs from what is observed in LASV-L, LACV-L, and Influenza D in which the 3' secondary binding site is already open at pre-initiation, with the 3' vRNA having access to it (Supplementary Fig. 14b)[8,13,27]. HTNV-L more closely resembles bat influenza A that displays a closed secondary 3' binding site at pre-initiation that opens at elongation to enable the binding of the 3'

**Fig. 6 | Conformational changes of HTNV-L between pre-initiation and elongation. a** Cut-away view cartoon representation of HTNV-L regions that are changing in conformations between pre-initiation and elongation. On the left and middle panels, the RNA are shown as cartoon and labeled. The bridge loop, the thumb-ring β-hairpin, the residues 1399–1402 and 1341–1344 are respectively colored in green, dark blue, gray, and purple. The other elements are colored as in Fig. 1. The bridge loop position is surrounded with a dotted line. On the right panel, superimposition with the pre-initiation colored in gray and the elongation colored as in the middle panel. For clarity, the RNA is not shown. Movements are indicated with arrows. Distances between the pre-initiation model and the elongation model are indicated. **b** Surface representation of HTNV-L in the pre-initiation and the elongation conformations. The RNA is shown as cartoon. The template exit tunnel and the 3′secondary binding site entrance are indicated with white and black dotted lines respectively. **c** Zoom on the 3′ secondary binding site at pre-initiation and elongation. 3′vRNA$_{1-25}$ is shown as yellow stick when present, domains are colored as in Fig. 1. The lower jaw reorganization and change of orientation at elongation is visible. Movements of the lower and upper jaws are depicted. On the right panel, superimposition with the pre-initiation colored in gray and the elongation colored as in the middle panel. For clarity the RNA is not shown. Movements are indicated with arrows. Distances between the pre-initiation and the elongation models are indicated. **d** Surface view of the 3′vRNA secondary site. Nucleotides and interacting domains are labeled. **e** Cartoon representation of the 3′vRNA secondary site. Two sub-panels are used to show residues that interact with the 3′vRNA$_{1-25}$ end. Nucleotides are numbered, and interacting residues are labeled. Hydrogen bonds are indicated as dotted lines.

template following its replication[26] (Supplementary Fig. 14c). Movements of bat influenza A and HTNV-L between pre-initiation and elongation are rather comparable. However, whereas only local movements of specific residues are involved in the opening of the 3′ RNA secondary binding site in Influenza, a more drastic reorganization and rotation of the lower jaw is observed in HTNV-L.

Altogether, the structures presented here describe the molecular basis of hantavirus replication. They depict the radical conformational rearrangements required for transitioning in a well programed and sequential manner between different key steps of this process, namely RNA recognition, replication pre-initiation, and replication elongation. We describe the structural key features and amino acid residues determining each state and transition, thereby providing ground for antiviral research by exposing potential targets for blocking the RNA synthesis by small molecules as, for instance, stabilizing the apo inactive form of HTNV-L. The structure-based biochemical assays we present here represent a platform for the screening of antivirals specific against hantaviruses, setting up the rational basis for implementing these assays in other systems including many highly pathogenic bunyaviruses. It paves the way towards future investigations of HTNV-L replication in the context of viral ribonucleoprotein assemblies and cell host interactors.

## Methods

### Cloning, expression, and purification
A construct containing the full-length *HTNV-L* gene (strain 76–118/Korean hemorrhagic fever, GenBank: X55901.1 UniProt: P23456) flanked in its N-terminus with a hexa-histidine tag was cloned in a pFastBac vector between NdeI and NotI restriction sites. A D97A mutation was introduced using the QuickChange II Site-Directed Mutagenesis method (Agilent). PCR primers were 5′-TTC AAA ATG ACC CCC GCT AAC TAC AAG ATC TCT GGC-3′ (forward) and 5′-GCC AGA GAT CTT GTA GTT AGC GGG GGT CAT TTT GAA-3′ (reverse). Reactions were performed with CloneAmp™ HiFi PCR Premix (Takara bio). To digest the parental and nonmutated dsDNA plasmid, the amplification reaction, previously analyzed by 1% agarose gel, was incubated with DpnI (NEB) at 37 °C for 1 h. The construct was sequenced (Genewiz) before further use.

The expressing baculovirus was prepared using the Bac-to-Bac method (Invitrogen)[28]. *Trichoplusia ni* High 5 cells (Invitrogen PN/51-4005 lot 1783124) were infected at $0.7 \times 10^6$ cells/mL with 0.1% v/v of baculovirus and harvested between 96 and 120 h after infection. Culture medium was centrifuged at $1000 \times g$ for 15 min and the cell pellets were frozen in liquid nitrogen.

To purify HTNV-L for structural characterizations, the cell pellets were resuspended in lysis buffer (30 mM HEPES pH8, 300 mM NaCl, 10 mM Imidazole, 2 mM TCEP, and 5% glycerol) supplemented with cOmplete EDTA-free protease inhibitor complex (Roche) and ribonuclease A (Roche). The lysate was sonicated during 3 min 30 s (10 s ON, 20 s OFF, and 40% intensity) and centrifuged during 45 min at $20,000 \times g$ and 4 °C. The supernatant was filtered at 0.8 μm and used

to purify HTNV-L$_{D97A}$ by nickel ion affinity chromatography (GE Healthcare). A washing step including the lysis buffer supplemented with 30 mM Imidazole was followed by the elution in 30 mM HEPES pH 8, 300 mM NaCl, 500 mM Imidazole, 2 mM TCEP, and 5% glycerol. HTNV-L$_{D97A}$ fractions were slowly diluted by two at 4 °C using the equilibration heparin buffer (30 mM HEPES pH8, 300 mM NaCl, 2 mM TCEP and 5% glycerol) and subsequently loaded to a heparin column (GE Healthcare). A washing step in 30 mM HEPES pH 8, 360 mM NaCl, 5 mM TCEP and 5% glycerol was followed by an elution in 30 mM HEPES pH 8, 500 mM NaCl, 5 mM TCEP and 5% glycerol. A final gel filtration step using a S200 size exclusion chromatography column (GE Healthcare) was done in GF buffer (30 mM HEPES pH 8, 250 mM NaCl, 5 mM TCEP). The best fractions were pulled and stored at −80 °C or directly used for cryo-EM.

Company names and catalog numbers of commercial reagents are indicated in Supplementary Data 2.

### EMSA assay
Interactions between HTNV-L and vRNA were evaluated by EMSA using fluorescently labeled RNAs. The RNA sequences are the following: 5′ vRNA$_{1-25}$ (5′- UAG UAG UAG ACA CCG CAA GAU GUU A-3′ – FAM), 3′ vRNA$_{1-25}$ (3′- AUC AUC AUC UGA GGC GUU UUC UUU C-5′ – CY5) and 5′ mut (5′- UAG GAG UAU CCA CCG CAA GAU GUU A-3′ – FAM). The unspecific RNA sequence used is FAM – 5′ - GUU UUG UAG AUA GGA GUA CAC UAC U-3′. RNAs at 0.2 μM were mixed with 2.5 μM of purified HTNV-L in binding buffer (30 mM TRIS-HCl pH 7.5, 5% glycerol, 500 mM NaCl, 10 mM TCEP, 1 mM EDTA) supplemented with 1 U/μl RNAse inhibitor (Roche). Poly-A at 0.16 μg/μl (Sigma) was included where indicated. Binding assays were done at RT for 10 min.

For the study of the 5′vRNA$_{1-25}$ binding as a single-stranded hook, RNA/DNA constructs were annealed and subsequently used in EMSA experiments. Reactions were stopped by adding native loading buffer (0.05 mM Bromophenol blue, 1 mM EDTA, 6% [v/v] glycerol) before separation on a 4% agarose gel. Native migration of RNA-Protein complex was performed at RT in 0.5× TAE (40 mM Tris-acetate, 1 mM EDTA) at 100 V for 30 min. Gels were visualized with a Phosphorimager Typhoon system and analyzed with ImageQuant TL program (Amersham).

### Activity assay
In vitro RNA synthesis was studied by incorporation of radiolabeled $^{32}$P-GTP in RNA product. For de novo replication assay, 0.6 μM of HTNV-L$_{D97A}$ was incubated with 6 μM of 5′vRNA$_{1-25}$ or 5′mut in a buffer containing 50 mM Tris pH 8, 250 mM NaCl and 5 mM β-mercaptoethanol for 30 min at 4 °C. 6 μM of 3′vRNA$_{1-25}$ was then added for 30 min at 4 °C. Reactions were started by adding 0.5 mM UTP/CTP/ATP, 0.04 mM GTP, 0.2 μCi/μl [$^{32}$P-GTP] and 5 mM MgCl$_2$. For replication in the absence of primer on Fig. 3b, lane 2, and Fig. 3c lanes 3 and 4, the [$^{32}$P-GTP] was 0.6 μCi/μl to boost the signal. When indicated, 0.5 mM dinucleotides (UpA, ApG, or GpU) were added to the mix. Reactions were incubated at 30 °C for 4 h.

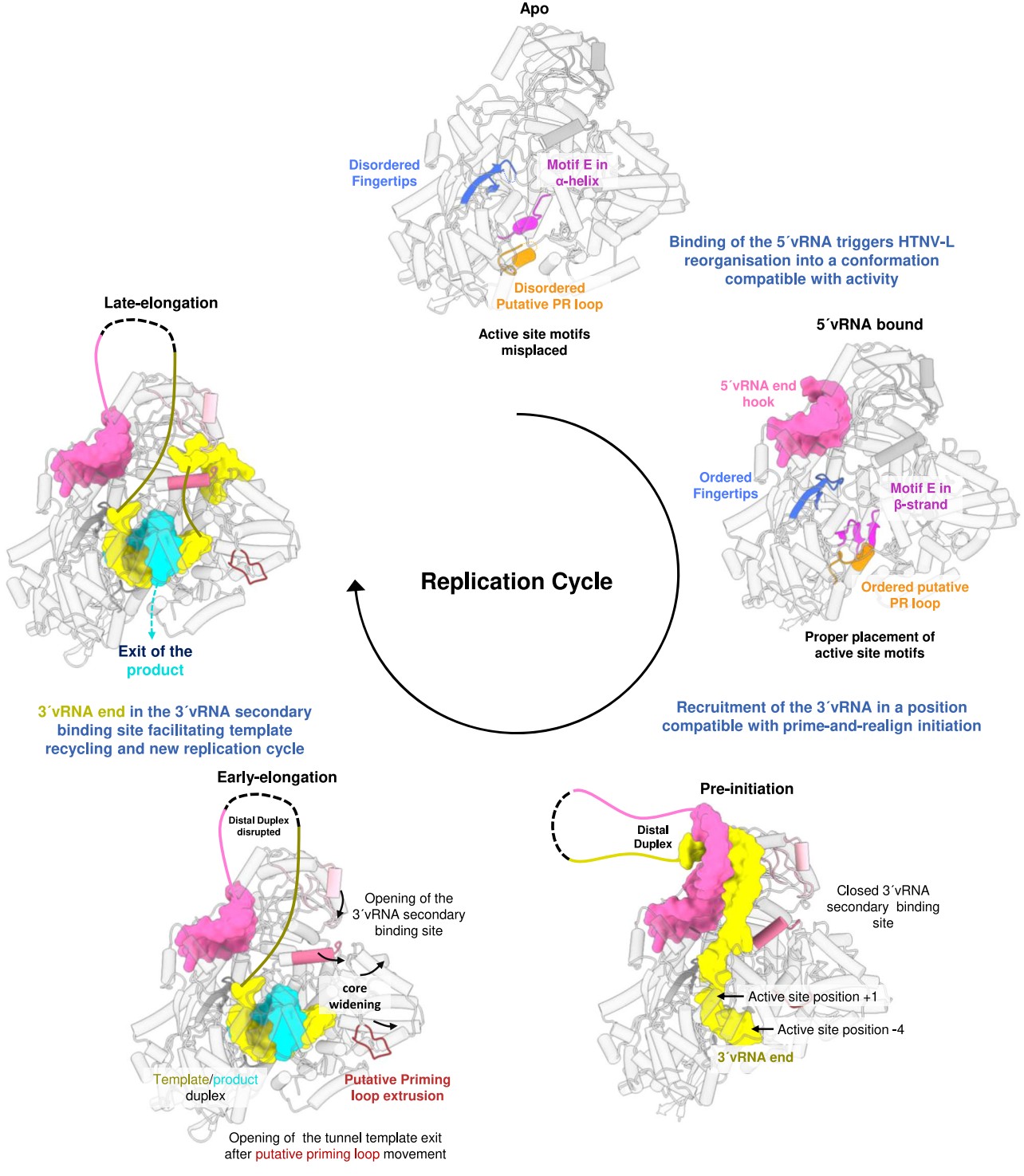

**Fig. 7 | Structure-based model of the mechanisms underlying HTNV-L replication.** Mechanistic model based on the structures. For the apo, the 5′vRNA-bound and the pre-initiation state, structures are used. For the early- and late elongation states, the elongation structure is used and only the RNA that we propose to be present in each state are shown. The 5′vRNA$_{1-25}$ (or the mutated 5′vRNA) and 3′ vRNA$_{1-25}$ are respectively colored in pink and yellow and shown as surface. HTNV-L structures are displayed as cartoon with the fingertips in blue, the motif E in magenta, the putative prime-and-realign (PR) loop in orange, the putative priming loop in dark red, the upper jaw in light pink, and in lower jaw in pink. In the pre-initiation state, the position +1 and −4 of the active site are indicated.

The dinucleotide-primed reactions were subsequently treated with 5 units of T4 polynucleotide kinase (NEB) at 37 °C during 10 min. This was performed to avoid difference of migration that could result from the difference of 5′-phosphorylation between the molecular weight markers (5′ monophosphorylated by T4 polynucleotide kinase for radiolabeling) and the dinucleotide-primed HTNV-L replication reaction (non-phosphorylated due to the 5′OH on the dinucleotide primer).

The unprimed replication products were further subjected to a digestion with Terminator Exonuclease (Lucigen) to study the presence of a 5′ triphosphate or 5′ monophosphate. The replication reaction was conducted as described above, stopped by incubation at 70 °C for 5 min to inactivate HTNV-L, cooled down 5 min at 4 °C and then incubated with 1U of Terminator Exonuclease for 1 h at 30 °C in the Terminator Exonuclease buffer.

All reactions were stopped by adding 2× volume of loading buffer (95% formamide, 1 mM EDTA, 0.025% SDS, 0.025% bromophenol blue, 0.01% xylene cyanol). The products were denatured by heating to 95 °C for 5 min and loaded on a 20% Tris-Borate-EDTA (TBE)−7M urea-polyacrylamide gel. The RNA products were separated with 1× TBE buffer for 2 h. The gels were exposed on a storage phosphor screen and read with an Amersham Typhoon scanner.

The molecular weight markers are (i) the 25-mer specific product that is complementary to the 3′vRNA (5′-UAG UAG UAG ACU CCG CAA AAG AAA G3′) and (ii) the 9-mer specific product that is complementary to nucleotides 2 to 10 of the 3′vRNA (5′-AGU AGU AGA-3′). The 25-mer and the 9-mer MW RNAs were radiolabeled in 5′ by incubation of the MW RNA at $10\,\mu M$ with 5 units of T4 polynucleotide kinase (NEB) in its buffer and $0.5\,\mu Ci/\mu l$ $[\gamma^{-32}P]$ ATP at 37 °C during 10 min. Reactions were stopped by incubation at 70 °C for 10 min and addition of loading buffer. The Decade marker (Thermofisher scientific) is also shown and was prepared according to manufacturer's instructions.

## Electron microscopy

To obtain the structure of HTNV-L$_{D97A}$ in apo conformation, $1\,\mu M$ of HTNV-L$_{D97A}$ in the GF buffer was incubated with 0.001% of glutaraldehyde (Sigma-Aldrich) during 15 min at 4 °C. 2 mM of MS(PEG)8 (ThermoFisher) was then added to the mix during 20 min at 4 °C. Cryo-EM grids were then immediately prepared.

To obtain the structure of HTNV-L$_{D97A}$ bound to 5′vRNA$_{1-25}$, $1.2\,\mu M$ of HTNV-L$_{D97A}$ in the GF buffer was incubated during 30 min at 4 °C with $12\,\mu M$ of 5′vRNA$_{1-25}$. 5 mM MgCl$_2$ and 0.001% glutaraldehyde were added to the mix for 15 min at 4 °C. Finally, 2 mM MS(PEG)8 was added during 20 min at 4 °C. Cryo-EM grids were then immediately prepared.

To obtain the structure of HTNV-L$_{D97A}$ in replication pre-initiation, $1.2\,\mu M$ of HTNV-L$_{D97A}$ in GF buffer was incubated with $12\,\mu M$ of 5′mut during 30 min at 4 °C. $12\,\mu M$ of 3′vRNA$_{1-25}$ was then added to the mix during 30 min at 4 °C. Subsequently, 0.001% glutaraldehyde was added to the mix during 15 min at 4 °C. Finally, the mix was supplemented with 2 mM MS(PEG)8 during 20 min at 4 °C. Cryo-EM grids were then immediately prepared.

To obtain the structure of HTNV-L$_{D97A}$ in replication elongation, the mix was prepared as follows: $1.4\,\mu M$ of HTNV-L$_{D97A}$ in the GF buffer was incubated (i) firstly with $14\,\mu M$ of 5′mut during 30 min at 4 °C, (ii) secondly with $14\,\mu M$ of 3′vRNA$_{1-25}$ during 30 min at 4 °C, (iii) thirdly with ATP, UTP, GTP, ApG to a final concentration of 500 μM and MgCl$_2$ to a final concentration of 5 mM. The replication reaction was run for 4 h at 30 °C. Subsequently, the replication mix was supplemented firstly with 0.001% glutaraldehyde for 15 min at 4 °C and secondly with 2 mM MS(PEG)8 during 20 min at 4 °C. Cryo-EM grids were then immediately prepared.

All conditions were frozen on UltraAuFoil 300 mesh, R1.2/1.3 grids (Quantifoil). The grids were glow-discharged at 25 mA during 45 s. 3.5 μL of sample were deposited on the grid that was subsequently blotted for 3 s (blot force 1) at 100% humidity and 4 °C in a Vitrobot Mark IV (Thermo Fisher Scientific) before plunge-freezing in liquid ethane.

Cryo-EM data collections were performed on a 200 kV Glacios cryo-TEM microscope (Thermo Fisher Scientific) equipped with a K2 summit direct electron detector (Gatan). Coma and astigmatism correction were performed on a carbon grid. Automated multi-holes (3 × 3) data collection was performed with SerialEM[29]. Movies containing 50 frames were acquired with a defocus between −0.8 μm and −2.2 μm at a nominal magnification of ×36,000 with pixel size of 1.145 Å. Total exposure dose was 50 e$^-$/Å$^2$.

## Image processing

Equivalent image processing strategy was used for all the datasets. Movies were imported in Relion 4.0 in nine optic groups based the nine different beam shifts required for their acquisition. They were realigned using Motioncor2[30] by applying the gain reference and the camera defect corrections. The micrographs were imported into cryoSPARC 3.3.2[31] for the following steps. CTF parameters were determined using the "Patch CTF estimation (multi)" tool on the non-dose-weighted micrographs. Low-quality micrographs were manually removed thanks to the "Manually Curate Exposure" tool (Supplementary Figs. 2, 9, 11). The selected micrographs were subjected to an automated picking with the "Blob picker tool" designed to select particles with a diameter comprised between 100 and 190 Å. Picking parameters were adjusted to increase the selectivity using the "Inspect particle picks" tool (Supplementary Figs. 2, 9, 11). The selected particles were extracted in a box size of 260 × 260 pixels$^2$ and binned twice. Two rounds of 2D classifications were applied to remove contaminants and low-quality particles (Supplementary Figs. 2, 9, 11). Selected particles were re-centered and re-extracted in a box size of 260 × 260 pixels$^2$ without any binning and used to generate the first initial model ("Ab Initio Reconstruction" tool). This initial model was used as a model for the first 3D reconstruction with all the particles selected ("Non-Uniform Refinement" tool). The following steps were performed in Relion 4.0[32,33]. A first round of 3D classification was performed with a global angular search. A circular mask of 180 Å diameter was applied at this stage. Classes showing HTNV-L$_{D97A}$ core were merged and used for 3D refinement with a 180 Å diameter circular mask. A second 3D classification with local angular search and a 180 Å diameter circular mask was performed. Particles that belonged to high-resolution 3D classes showing the core (and a template/product duplex RNA density for HTNV-L$_{D97A}$ replication elongation) were selected and used for a masked 3D refinement with local angular search. CTF refinement, estimation and correction of asymmetrical and symmetrical aberrations, correction of anisotropic magnification were performed. The resulting particles were subjected to a 3D masked refinement with local angular search. Bayesian polishing was used for per-particle reference-based beam-induced motion correction. Finally, a 3D refinement with local angular search was performed, followed by post-processing. For each final map, the global resolution is based on the FSC 0.143 cutoff criteria. Local resolution variation of maps is also estimated using Relion 4.0 (Supplementary Fig. 2, 9, 11).

## Model building in the cryo-EM maps

The filtered local-resolution maps were used to build all the models manually using COOT[34]. An AlphaFold model[35] comprising the thumb and the thumb-ring region was used to guide the model building in COOT of parts of the thumb-ring (in areas where the map density was present but its quality was too low for ab initio unambiguous building). Each model was refined using Phenix real-space refinement[36]. Atomic model validation was performed with Molprobity[37] and the PDB validation server. Model resolution according to the cryo-EM maps was estimated with Phenix at the 0.5 FSC cutoff. Figures were created using ChimeraX[38] and Chimera[39]. PDB2PQR[40] and APBS[41] were used to calculate the electrostatic potential. PISA[42] and PLIP[43] were used to analyze protein-RNA interactions. Mole2.0 was used to visualize the tunnels in HTNV-L core[44].

## Multiple alignment

Multiple alignment was done in Muscle[45] and displayed with ESPript 3.0[46].

## Statistics and reproducibility

Expression and purification of HTNV-L$_{D97A}$ was repeated 10 times and provided similar results as Supplementary Fig. 1a. EMSA assays were repeated 3 times and provided similar results as Fig. 2a, b. Activity assays were repeated 3 times and provided similar results as Fig. 3a–c.

## Reporting summary

Further information on research design is available in the Nature Portfolio Reporting Summary linked to this article.

## Data availability

The coordinates and structure factor generated in this study have been deposited in the Protein Data Bank and the Electron Microscopy Data Bank database under accession codes: 8C4S, EMD-16427 (HTNV-L apo core), 8C4T, EMD-16428 (5′vRNA-bound HTNV-L), 8C4U, EMD-16429 (HTNV-L in pre-initiation state), 8C4V, EMD-16430 (HTNV-L in elongation state). Source data are provided with this paper. Access to the "minimum dataset" that is necessary to interpret, verify and extend the research in the article can be made available through request to H.M. and J.R. Source data are provided with this paper.

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

## Acknowledgements

We thank Guy Schoehn for setting up the IBS/ISBG EM platform and for discussion; Eleftherios Zarkadas for support and technical advice on cryo-EM data collection; Martin Pelosse for technical advice on expression; Aymeric Peuch for setting up and maintaining the EM computing cluster; Barbara Selisko and Delphine Baud for management of the radioactivity facilities; Dominique Housset, Allison Ballandras-Colas, Florian Chenavier, Stephen Cusack, and Thibaut Crépin for useful discussions; Claire Debarnot, Daphna Fenel, and Madalen Le Gorrec for technical support; Mackensie Thompson for English proofreading. This work used the platforms of the Grenoble Instruct-ERIC center (ISBG; UAR 3518 CNRS-CEA-UGA-EMBL) within the Grenoble Partnership for Structural Biology (PSB) and the Marseille platform for integrative structural biology (AFMB-PBSIM platform), both supported by FRISBI (ANR-10-INBS-05-02). The ISBG; UAR 3518 was also supported by GRAL, financed within the University Grenoble Alpes graduate school (Ecoles Universitaires de Recherche) CBH-EUR-GS (ANR-17-EURE-0003). The electron microscope facility is supported by the Auvergne-Rhône-Alpes Region, the Fondation pour la Recherche Médicale (FRM), the fonds FEDER and the GIS-Infrastructures en Biologie Santé et Agronomie (IBiSA). We acknowledge the European Synchrotron Radiation Facility for the provision of beam time on CM01 and we would like to thank Grégory Effantin for their assistance. We thank the platform staff that enabled us to perform these analyses. IBS acknowledges integration into the Interdisciplinary Research Institute of Grenoble (IRIG, CEA). This work was supported by the ANR-19-CE11-0024 to H.M. and J.R. and the Institut Universitaire de France endowment to H.M. S.B.G. was supported by a Ph.D. fellowship from the Fondation pour la Recherche Médicale (FRM) ECO201806006831 and FDT202106013088.

## Author contributions

S.B.G. performed the cloning and site-directed mutagenesis. S.B.G., B.A., and Q.D.T. expressed and purified HTNV-L. S.B.G. performed EMSA assays. Q.D.T. and S.B.G. performed replication assays. Q.D.T. and B.A. optimized the cryo-EM conditions. H.M. and Q.D.T. collected cryo-EM data on a Thermo Fischer Scientific Glacios EM. Q.D.T. and B.A. performed cryo-EM image processing. B.A. determined the first cryo-EM reconstruction of HTNV-L. Q.D.T. determined all the deposited structures. B.A., H.M., and Q.D.T. built the models based on the cryo-EM maps. H.M., Q.D.T., and B.A. performed structural analysis. H.M. supervises Q.D.T. and B.A.; J.R. supervises S.B.G. The project was conceived by H.M. and J.R. This project used funding obtained by H.M. and J.R. The manuscript was written by H.M. and Q.D.T. with input from all the authors.

## Competing interests

The authors declare no competing interests.
