## [Peer Review File · Nature Communications]

Structures of active Hantaan virus polymerase uncover the mechanisms of Hantaviridae genome replicationReviewer #1 (Remarks to the Author):

Hantaviruses are negative-stranded RNA viruses that cause life-threatening zoonotic infections in humans. Hantaan viral RNA synthesis is the key to the viral replication cycle and is done by the virally encoded multi-functional RNA-dependent RNA-polymerase. Trouilleton et al. described the apo and elongation structures of the Hantaan virus polymerase core and provided conditions for in vitro replication activity. The authors highlighted domain reorganization and activation upon the template/product binding.

Overall, it is a well-written paper with solid experimental data. This paper reported the cryo-EM structures of apo-form and RNA-bounded HTNV L complex, which provide insights into the mechanisms of Hantaviridae RNA synthesis. The data unambiguously showed the conclusion. However, there are still some minor issues in this manuscript that need to be addressed:

1. Line 105-108 should refer to the electrostatic potential surface figure for better illustration.
2. The in vitro replication assays part. Is the chemical group at the 5' end of the primer ApG "-OH"? As the chemical group of the RNA will influence its migration in RNA denaturing gel. Especially here, the 9-mer position band in Figure 3b may be a fusion band including RNA products from 5-9 nts. The authors can easily confirm this by treating the RNA products with T4 PNK.
3. In lane 9 of Figure 3b, some bands around 12 nts also were observed in the gel. What are those bands?
4. What is the difference between lane 1 and 2 in Figure 3b except for the concentration of [32P-GTP]? Why lane 1 is nothing while lane 2 generates RNA products?
5. Figure 1b, there has one semicircular helices, which may be two helices.
6. Figure 5a, what is the function of the secondary binding site?
7. Line 186, should it be "followed by"?
8. Line 431,434 and 435, the unit of temperature 30°C, 70°C, and 4°C.
9. Line 471, "during" or "for"

Reviewer #2 (Remarks to the Author):

The stated goal of this manuscript by Trouilleton et al. was to solve the structures of Hantaan virus polymerase at different stages of replication. The authors succeed in solving 5 structures at better than 3.5 Å resolution. This manuscript provides many high quality structures of a viral polymerase of importance to clearly interpret differences between states and which will provide a platform for the development of small molecules targeting the replication cycle of hantaviruses. However, it is unclear if the primary finding of an "inactive" conformation of motif E is merely a dependence on magnesium and/or the effects of crosslinking and the manuscript could benefit from better figures to more clearly show differences that the authors identify and a more concise body of text.

Major points:

Much of results subsection: "Structure determination of HTNV-L core" seems extraneous and would likely be easier to read if comparisons were instead made to existing structures of other Bunyavirales polymerases given that a statement about differences is present at the end of the introduction. Perhaps much of the results discussing specific sub-domains and contacts could be moved to supplemental text to improve the readability of the manuscript.

The authors use of glutaraldehyde to crosslink their complexes is concerning given that conclusions are made based on differences between structures that could be due to the

effects of crosslinking. Could the authors please comment on why they chose to do this?

Figure 2/Supp Figure 4: Much is written in the text regarding conformational changes depicted in these figures. These are impossible to tell with the figures as currently shown. Overlays of the two structures with a common portion aligned with arrows and possibly distances to show the direction and magnitude of these changes would help significantly.

It's unclear what is causing the conformational changes between the apo and 5'vRNA-bound structures. The presence of magnesium ions in the 5'vRNA-bound structure but not the apo structure could explain these differences. The methods section does not list magnesium ions as a component in any buffer but could be present in the purified RNA that is added. What does the structure of the apo complex in the presence of magnesium ions look like? The use of "active" and "inactive" when referring to the 5'vRNA-bound and apo structures is also misleading as HNTV-L bound to 5'vRNA is not active in itself.

Line 162-170: The authors claim that mutation of the vRNA was necessary to avoid dsRNA formation. We are unconvinced that the resulting structures are identical to an un-mutated complex. In fact, the EMSA in Fig 3a indicates that the mutated vRNA binds potentially tighter than the native sequence. In addition, the 5'WT sequence did not generate any replication product which is very concerning. This does not appear to be a universal problem for all Bunyavirales polymerases. Have the authors considered assembling these complexes in a step-wise manner (first the 5'vRNA followed by the 3'vRNA) to prevent premature complementary binding of the vRNA strands to each other? Lines 223-224 say the 5' mutant binds like WT but it is unclear how this is concluded without a structure of the WT complex.

There are many inconsistent capitalization and naming conventions throughout the manuscript. For example: hantavirus (line 39) vs Hantaviruses (line 42), 5'vRNA1-25 (line 116) vs 5'vRNA-bound (line 135) vs 5'end (line 161) to name a few. Consistent, specific naming would improve the readability of the manuscript.

Figure 6: Superimposition of these structures would more clearly illustrate the authors points than a side-by-side comparison.

Figure 7: Increase size of text, there is much white space that can be filled. To be a mechanism diagram, arrows should indicate the path between states, the surface view and cartoon view are redundant, consider just using the cartoon view so that each panel can be enlarged.

In our opinion, much of the discussion comparing the HNTV-L structures to other Bunyavirales polymerases could be moved to the results and replace the text we suggest moving to supplemental text. At a minimum, supplemental figures comparing these structures would be of high value to many readers of this manuscript.

Minor points:

line 60-62: "polymerase surface" is very vague and uninformative.

Results sub-section titles are inconsistent in theme ("5'vRNA allosterically activates the active site through major conformation changes" ← great! vs "In vitro replication assays" ← needs work)

line 122: It's unclear what the authors intended with the statement that EMSA assays suggest that 5'vRNA binds as a single-stranded hook. EMSA assays only indicate whether binding occurs, not the nature of that binding interaction.

Line 172: majoritarian is not a word

Lines 191-194: These sentences don't seem necessary to include.

Lines 202-208: This paragraph could benefit from reorganization. Explaining which 3 nucleotides were used and the expected result prior to the actual results would read much better.

Line 210: It is unclear what "they" refers to.

Supp Fig. 7 would benefit from the same changes as Fig. 2/Supp. Fig. 4.

Lines 268-274: Much of this seems unnecessary and could be moved to supplemental text.

Line 318: 5' → 5'vRNA

Line 349: infirming → disproving?

Figure 1: text is too small (in a.), some colors as well

Figure 2: a,b) Why does the size of the EMSA products shift down in the presence of poly(A)? This seems unexpected.

b) I'm very confused by the ss and ds labels. How is that validated?

Figure 3: b) MW25 ladder is very difficult to interpret. Consider redoing these experiments with a different ladder that is clearer to interpret.

Figures 4&5: Yellow text is very difficult to read, add a black border around the text. Same with most of the base labels in 4c, 5d.

Reviewer #3 (Remarks to the Author):

The last couple of years, the RNA polymerases of various negative sense RNA viruses have been extensively studied using crystallography and cryo-EM. These studies revealed the various steps of the transcription or replication cycle, and identified many shared principles. Here, Trouilleton et al present the near-complete replication cycle, split over 5 cryo-EM structures, of a hantavirus RNA polymerase. The authors employ biochemical experiments and previously explored promoter mutations to study the initiation and elongation activities of the hantavirus RNA polymerase in vitro. Overall, the experiments are well-designed, the structures insightful and carefully annotated, and the manuscript, with the exception of a handful of grammatical errors, very clearly written. While other studies provide insight into many aspects of the nucleotide addition cycle of negative sense RNA virus RNA polymerases and in particular the transcription cycle, this study is a tour-the-force on the hantavirus RNA polymerase replication cycle. The data reported here will be useful for our fundamental understanding of negative sense virus replication, the appreciation of unique characteristics of different viral replication complexes (some of which the authors identify for the hantavirus RNA polymerase), and the development of broad-spectrum as well as virus-specific antivirals going forward. I don't have any substantive concerns about this manuscript.

We thank you for your positive appreciations and constructive comments on our manuscript entitled "Structures of active Hantaan virus polymerase uncover the mechanisms of *Hantaviridae* genome replication". Please find below our point-by-point responses.

Reviewer #1 (Remarks to the Author):

Hantaviruses are negative-stranded RNA viruses that cause life-threatening zoonotic infections in humans. Hantaan viral RNA synthesis is the key to the viral replication cycle and is done by the virally encoded multi-functional RNA-dependent RNA-polymerase. Trouillette et al. described the apo and elongation structures of the Hantaan virus polymerase core and provided conditions for in vitro replication activity. The authors highlighted domain reorganization and activation upon the template/product binding.

Overall, it is a well-written paper with solid experimental data. This paper reported the cryo-EM structures of apo-form and RNA-bound HTNV L complex, which provide insights into the mechanisms of Hantaviridae RNA synthesis. The data unambiguously showed the conclusion.

However, there are still some minor issues in this manuscript that need to be addressed:

Minor points:

1. Line 105-108 should refer to the electrostatic potential surface figure for better illustration. We have added the **Supplementary Figure 5a** that shows the electrostatic potential of the 5'vRNA binding site.

2. The in vitro replication assays part. Is the chemical group at the 5' end of the primer ApG "-OH"? As the chemical group of the RNA will influence its migration in RNA denaturing gel. Especially here, the 9-mer position band in Figure 3b may be a fusion band including RNA products from 5-9 nts. The authors can easily confirm this by treating the RNA products with T4 PNK.

We would like to thank Reviewer 1 for this comment that was really useful and constructive. To avoid difference of migration that could be due to the difference of 5'-phosphorylation between the molecular weight markers (5' monophosphorylated by T4 PNK for radiolabeling) and the dinucleotide-primed HTNV-L replication reaction (non-phosphorylated due to the 5'OH on the dinucleotide primer), we have treated all the HTNV-L dinucleotide-primed replication products with PNK. This is now indicated in the Material and Methods section:

"The dinucleotide-primed reactions were subsequently treated with 5 units of T4 polynucleotide kinase (NEB) at 37°C during 10 min. This was performed to avoid difference of migration that could result from the difference of 5'-phosphorylation between the molecular weight markers (5' monophosphorylated by T4 polynucleotide kinase for radiolabeling) and the dinucleotide-primed HTNV-L replication reaction (non-phosphorylated due to the 5'OH on the dinucleotide primer)."

We have in addition solved the problem of low-molecular weight RNA migration that was pointed out by Reviewer 1. We replaced the TCEP in HTNV-L buffer with β -mercaptoethanol

as we identified TCEP as being the cause of low-molecular weight blurring. The new gel provided in **Fig. 3b** can be unambiguously interpreted. For the reaction with 4 nucleotides no modifications were seen compared to our previous gel, thereby confirming the result description and interpretation. For the reaction with a 3-nucleotide subset, the gel confirms the presence of a 9-mer product as initially stated in the text and also identifies 6-mer and 11-/12-mer products that could correspond to un-realigned and two-times realigned products (see Reviewer 1 point 3).

We have updated the text as such:

“This nicely confirms the presence of the expected 9-mer product (**Fig. 3b**, lane 7). We also visualize ~6-mer and ~11-/12-mer products that could respectively correspond to un-realigned and two-times realigned products generated by imperfect prime-and-realign initiation *in vitro* (**Fig. 3b**, lane 7).”

3. In lane 9 of Figure 3b, some bands around 12 nts also were observed in the gel. What are those bands?

In all replication reactions, we observe the expected product as a main product and an additional longer product is visible that we estimate being around 3 nucleotides longer. For lane 7 of **Fig. 3b**, as pointed by the Reviewer, we observe a ~11-/12-mer product in addition to the 9-mer main product. We have already visualized this behavior in replication reactions of La Crosse Orthobunyavirus polymerase (Arragain *et al*, Nature communications, 2022). RNA next-generation sequencing had identified the addition of 3 nucleotides at the 5' end of the product, that were corresponding in sequence to an extra triplet repeat at the 5' end. This can be explained by two-rounds of prime-and-realign replication initiation.

We have added two sentences in the main text to mention this interesting behavior:

“We also visualize faint product elongated by ~3 nucleotides (~27-mer above the 24-mer main product on line 5, ~26-mer above the 23-mer product on line 6). These faint products could correspond to a double prime-and-realign reaction that would result in an extra-triplet incorporation at the 5'-end of the product.”

“We also visualize ~6-mer and ~11-/12-mer products that could respectively correspond to un-realigned and two-times realigned products generated by imperfect prime-and-realign initiation *in vitro* (**Fig. 2**, lane 9).”

4. What is the difference between lane 1 and 2 in Figure 3b except for the concentration of [³²P-GTP]? Why lane 1 is nothing while lane 2 generates RNA products?

The concentration of [³²P-GTP] is indeed the only difference between lanes 1 and 2 of **Fig. 3b**. A very faint signal exists in lane 1 that requires using the different threshold for gel visualization (see **Rebuttal letter Figure 1**, displaying the gel of **Fig. 3b** at a different threshold). We decided to keep lane 1 in the Figure 3b gel so that the reader could compare lane 1 with the other bands of the gel done using the same hot GTP concentration, therefore clearly showing the product obtained without dinucleotide primer is very faint compared to the products obtained with dinucleotide primers. To better visualize the product in the absence of dinucleotide primer, we multiplied by 3 the concentration of [³²P-GTP] in lane 2. This increases the amount of radioactive GTP incorporated, thereby boosting the signal that becomes clearly visible.

This is indicated in the Material and Methods:

“For replication in the absence of primer on **Fig. 3b**, lane 2 and **Fig. 3c** lanes 3 and 4, the [³²P-GTP] was 0.6 μCi/μl to boost the signal.”

And we added a description in the main text:

“(Fig. 3b, lanes 1 and 2, [³²P-GTP] was multiplied by 3 in lane 2 compared to line 1 to boost the signal and clearly visualize the product)”.

Rebuttal letter Figure 1: Lanes 1 to 3 of Fig 3b shown at two different thresholds. On the right side, the threshold chosen allows to clearly visualize the lane 1 band but results in too high signal for the other lanes. On the left side, the threshold chosen allows to clearly visualize the other lanes.

5. Figure 1b, there has one semicircular helices, which may be two helices.

This has been corrected and now shows two helices.

6. Figure 5a, what is the function of the secondary binding site?

The 3'vRNA secondary binding site is likely to be linked to recycling of the 3'vRNA for subsequent replication cycle. This is now indicated at the end of the discussion:

“Sequestration of the 3'vRNA end in the 3'vRNA secondary binding site at late-elongation is likely to ensure efficient recycling of the 3'vRNA template for future rounds of replication.”

7. Line 186, should it be “followed by”?

Yes, thank you, this has been corrected.

8. Line 431,434 and 435, the unit of temperature 30 °C, 70 °C, and 4 °C.

This has been corrected.

9. Line 471, “during” or “for”

Yes, a word was missing, this has been corrected.

Reviewer #2 (Remarks to the Author):

The stated goal of this manuscript by Trouilleton et al. was to solve the structures of Hantaan virus polymerase at different stages of replication. The authors succeed in solving 5 structures at better than 3.5 Å resolution. This manuscript provides many high quality structures of a viral polymerase of importance to clearly interpret differences between states and which will provide a platform for the development of small molecules targeting the replication cycle of hantaviruses. However, it is unclear if the primary finding of an “inactive” conformation of motif E is merely a dependence on magnesium and/or the effects of crosslinking and the manuscript could benefit from better figures to more clearly show differences that the authors identify and a more concise body of text.

We thank Reviewer 2 for his positive comments about the quality of the structures and the importance of the results. To answer Reviewer 2 point, we determined HTNV-L structure with magnesium in the absence of glutaraldehyde. The resulting structure does not differ from the apo structure (**Rebuttal letter Figure 2**), the active site including the motif E is in an inactive conformation and magnesium is not seen. This confirms that: (i) glutaraldehyde does not induce an inactive conformation, (ii) that the presence of magnesium is not sufficient to switch from an “inactive” to an “active” conformation.

Rebuttal letter figure 2:

a global view of the HTNV-L structure done in presence of 5mM $MgCl_2$. The apo structure is fitted. HTNV-L is colored as in **article Fig. 1**.

b zoom on motif E that is clearly alpha-helical.

c zoom on the active site colored as in **article Fig. 2f**. On the left side, the apo model is displayed in the map and fits properly. The residues D972, D1099 and E1170 are not properly positioned to bind a Mg^{2+} ion. On the right side, the HTNV-L-5'vRNA-bound/ Mg^{2+} -bound model is displayed and clearly does not fit the map (the motif E in magenta does not fit, the putative PR-loop in orange does not fit, no density for the Mg^{2+} is visible).

Major points:

1. Much of results subsection: “Structure determination of HTNV-L core” seems extraneous and would likely be easier to read if comparisons were instead made to existing structures of other *Bunyavirales* polymerases given that a statement about differences is present at the end of the introduction. Perhaps much of the results discussing specific sub-domains and contacts could be moved to supplemental text to improve the readability of the manuscript.

We have significantly modified the “Structure determination of HTNV-L core” subsection to include comparisons with other *Bunyavirales* polymerases and with Influenza polymerase. We have also added the following paragraph:

“If the global domain organization is conserved with other sNSV polymerases, two neighboring surface-exposed structural additions are specific to *Hantaviridae* polymerase: (i) a two β -strand insertion (residues 949 and 963) that corresponds to the α -helical Californian insertion observed in LACV-L and (ii) a two α -helix insertion between residues 1109 and 1137 (**Supplementary Fig. 4b**). The functional importance of these insertions remains to be determined. On the contrary, the α -ribbon that protrudes from the finger domain in the other *Bunyavirales* polymerase structures, that interacts with the distal duplex and undergoes significant movements during activity, is absent in HTNV-L (**Supplementary Fig. 4c**).”

Two new **Supplementary figures 3 and 4** have been added to compare the structure of *Bunyavirales*.

We thank Reviewer 2 for this comment as the additional comparison requested improves to the article. Please however note that the comparisons that Reviewer 1 proposed us to add necessarily involves the description of sub-domains insertions and deletion. Indeed, the domains are conserved between *Bunyavirales* structures and the differences are within these sub-domains.

About description of the contacts between the RNA and the protein, much of them are shown in Figures (for example **Fig. 2d, Fig. 6e**) and are listed in **Supplementary Table 2**. We feel here the need to point our efforts so that the manuscript is as easy to read as possible. This is acknowledged by Reviewer 1 who says that it is a “well-written paper” and Reviewer 3 who states that “the manuscript is very clearly written”.

2. The authors use of glutaraldehyde to crosslink their complexes is concerning given that conclusions are made based on differences between structures that could be due to the effects of crosslinking. Could the authors please comment on why they chose to do this?

The glutaraldehyde was used at 0.001%, a very low concentration, in an attempt to stabilize the ENDO and the C-terminus of HTNV-L (and Reviewers might have requested its addition if we had not done it). It was added only after the interaction with RNA (for the 5'vRNA-bound map and the pre-initiation map) and after the replication reaction (for the elongation map). If glutaraldehyde was inducing conformational changes, all the structures would be trapped in the same conformation, which is not the case. Here, conformational changes between 5'vRNA-HTNV-L, pre-initiation and elongation are clearly seen as assessed by the presence of

the expected RNA. The conformations are typical for active states as seen in the comparison with LACV-L (Arragain *et al*, 2022), SFTSV-L (Williams *et al*, NAR, 2023) and Influenza (Wandzik *et al*, Cell, 2020), described in the discussion.

To be sure the conformational changes visualized between the apo and the 5'vRNA-bound HTNV-L map are not an effect of the cross-linking, we have determined two other structures in the absence of glutaraldehyde: (i) HTNV-L with magnesium and (ii) HTNV-L with 5'vRNA. These structures are not different from the (i) apo HTNV-L done in presence of 0.001% glutaraldehyde (**Rebuttal letter Figure 2**) and (ii) 5'vRNA-HTNV-L done in presence of 0.001% glutaraldehyde (**Rebuttal letter Figure 3**). This confirms the validity of the structures.

Rebuttal letter figure 3: superimposition of the 5'vRNA₁₋₂₅-bound map with 0.001% glutaraldehyde (left) and without glutaraldehyde (right).

a The 5'vRNA₁₋₂₅-bound HTNV-L model is fitted inside the maps. The 0.001% glutaraldehyde stabilizes a bit the thumb-ring but does not induce any conformational change.

b Zoom on the active sites of both maps. The active sites are identical in both cases. The motifs E are in the beta-strand conformation in both cases. The motifs are properly positioned to coordinate a magnesium ion for which specific density is clearly visible.

3. Figure 2/Supp Figure 5: Much is written in the text regarding conformational changes depicted in these figures. These are impossible to tell with the figures as currently shown. Overlays of the two structures with a common portion aligned with arrows and possibly distances to show the direction and magnitude of these changes would help significantly.

We thank Reviewer 2 for this comment that clarifies the figures.

We have added an overlay of the 2 structures on **Figure 2e**. Arrows indicating movements and distances have now been added.

Figure 2f and g have been simplified to clearly depict each element mentioned in the text. The addition of an overlay of **panels 2f and 2g** would have made the panels too crowded and too small, preventing clear visualization. We thus show the requested overlay in **Supplementary Fig 8**, that is supporting **panels 2f and 2g**:

- **Supplementary Fig 8a** shows the fingernodes, the finger, the fingertips and motif B. An overlay of the apo and the 5'-bound structure is shown.

- **Supplementary Fig 8b** shows the other active site motifs and the PR loop. Arrows indicating movements and distances are indicated.

Distances and direction of movements are now shown.

4. It's unclear what is causing the conformational changes between the apo and 5'vRNA-bound structures. The presence of magnesium ions in the 5'vRNA-bound structure but not the apo structure could explain these differences. The methods section does not list magnesium ions as a component in any buffer but could be present in the purified RNA that is added. What does the structure of the apo complex in the presence of magnesium ions look like? The use of "active" and "inactive" when referring to the 5'vRNA-bound and apo structures is also misleading as HNTV-L bound to 5'vRNA is not active in itself.

We have determined HTNV-L structure in presence of magnesium in the absence of glutaraldehyde (**Rebuttal letter figure 2**). The map obtained does not differ from the apo map clearly showing that it is not the magnesium itself that is causing the reorganization of the active site. The active site motifs are not in position compatible with activity.

We have removed the use of "active"/"inactive" when referring to the 5'vRNA/apo structure: "Comparative analysis of HTNV-L apo and HTNV-L 5'vRNA₁₋₂₅-bound map reveals unexpected conformational changes with a switch from a previously unknown apo structure incompatible with activity to a 5' vRNA₁₋₂₅-bound map with motif organization compatible with activity (**Fig. 2e**)."

5. Line 162-170: The authors claim that mutation of the vRNA was necessary to avoid dsRNA formation. We are unconvinced that the resulting structures are identical to an un-mutated complex. In fact, the EMSA in Fig 3a indicates that the mutated vRNA binds potentially tighter than the native sequence. In addition, the 5'WT sequence did not generate any replication product which is very concerning. This does not appear to be a universal problem for all Bunyavirales polymerases. Have the authors considered assembling these complexes in a step-wise manner (first the 5'vRNA followed by the 3'vRNA) to prevent premature complementary binding of the vRNA strands to each other? Lines 223-224 say the 5' mutant binds like WT but it is unclear how this is concluded without a structure of the WT complex.

There is a structure of the WT complex: HTNV-L bound to WT 5'vRNA. We have added a new **Supplementary figure 5b** that shows the binding modes of 5'WT and 5'mut. This clearly shows that 5'WT and 5'mut5 bind as a hook in the same pocket.

The fact that 5'WT sequence does not generate replication product *in vitro* has already been observed for LACV-L (Arragain *et al*, Nature communications, 2022). As HTNV-L and LACV-L are the most conserved within the *Bunyavirales* polymerase structures solved so far, this observation is therefore not unexpected.

Concerning the assembly of the complexes in a step-wise manner, first a 30 min incubation with the 5' and then addition of the 3', we have always done what Reviewer 2 is requesting, as indicated in the Material and Methods.

6. There are many inconsistent capitalization and naming conventions throughout the manuscript. For example: hantavirus (line 39) vs Hantaviruses (line 42), 5'vRNA1-25 (line 116) vs 5'vRNA-bound (line 135) vs 5'end (line 161) to name a few. Consistent, specific naming would improve the readability of the manuscript.

We have carefully checked the capitalization and naming convention through all the manuscript. We now refer to 5'vRNA₁₋₂₅, 3'vRNA₁₋₂₅ and 5'mut. When we need to specifically to the extremity of the RNA we kept 5'-end or 3'-end.

7. Figure 6: Superimposition of these structures would more clearly illustrate the authors points than a side-by-side comparison.

We now show superposition in addition to side-by-side comparison on **Fig. 6 a and c**.

8. Figure 7: Increase size of text, there is much white space that can be filled. To be a mechanism diagram, arrows should indicate the path between states, the surface view and cartoon view are redundant, consider just using the cartoon view so that each panel can be enlarged.

Thank you for this comment. We have taken into account all these suggestions to update **Figure 7** and we agree it clarifies the figure.

9. In our opinion, much of the discussion comparing the HTNV-L structures to other Bunyavirales polymerases could be moved to the results and replace the text we suggest moving to supplemental text. At a minimum, supplemental figures comparing these structures would be of high value to many readers of this manuscript.

An important aspect of a discussion is to summarize the results and outline their interpretation in light of the published literature. This is what we do when we discuss interesting aspects visualized in HTNV-L and compare them with not only *Bunyavirales* polymerases but also Influenza, Dengue and Zika polymerases.

Concerning which exact text should be moved to supplemental text, Reviewer 2 may refer to his main point 1, to which we have replied in point 1 answer.

We have added a new **Supplementary Figure 3** to compare the general structures of *Bunyavirales* polymerases determined so far and a new **Supplementary Figure 4** that shows the insertions and deletions compared to other *Bunyavirales* polymerase cores. This adds to

Supplementary Figure 13 that compares the putative PR loop and priming loop from different sNSV polymerases and **Supplementary Figure 14** that compares the 3' secondary binding sites from different sNSV polymerases.

Minor points:

1. *line 60-62: "polymerase surface" is very vague and uninformative.*

This has been updated:

"The 5' binds as a hook in a specific site located on the surface of the polymerase core region, whereas the 3' binds either in the active site or in a distinct 3'-end secondary binding site."

"Complementary nucleotides around positions 11 to 20 form a distal duplex that protrudes from the polymerase core".

2. *Results sub-section titles are inconsistent in theme ("5'vRNA allosterically activates the active site through major conformation changes" ← great! vs "In vitro replication assays" ← needs work)*

We thank Reviewer 2 for this remark. We have updated several sub-section titles, taking into account the 60 character maximum requested by Nature Communication:

"HTNV-L actively replicates 3' vRNA template *in vitro*"

"Pre-initiation state: 3'vRNA in position for prime-and-realign initiation"

"The 3'vRNA secondary binding site opens at elongation"

3. *line 122: It's unclear what the authors intended with the statement that EMSA assays suggest that 5'vRNA binds as a single-stranded hook. EMSA assays only indicate whether binding occurs, not the nature of that binding interaction.*

The text has been updated as such: "5'vRNA₁₋₂₅ binding through its single-stranded 5'-end was suggested by EMSA assays of the 5'vRNA₁₋₂₅ incubated with complementary DNA of various lengths (**Fig. 2b**). It identified 5'vRNA₁₋₂₅ binding only when a minimum of 10 nucleotides were left single-stranded on the 5'-end."

Indeed, the EMSA shown in Figure 2b shows that double-stranded RNA does not bind to HTNV-L and that binding to HTNV-L is seen when the 5' end of the 5'-vRNA is left single stranded.

4. *Line 172: majoritarian is not a word.*

This has been corrected as: "Faint products were observed with the main product being a 24-mer nucleotide product".

5. *Lines 191-194: These sentences don't seem necessary to include.*

We have deleted these sentences in the revised version.

6. *Lines 202-208: This paragraph could benefit from reorganization. Explaining which 3 nucleotides were used and the expected result prior to the actual results would read much better.*

The text has been updated as requested:

"We also tested if the replication could be stalled at early-elongation by incubating HTNV-L with a 3-nucleotide subset (**Fig. 3b**, lane 7). This is expected to result in the generation of a 9-nucleotide product that corresponds to nucleotides 2 to 10 of the template, as (i) ApG primer

corresponds to position 2 of the template and (ii) the absence of CTP in the mix induces a stop at position 11 of the template that is a G. This nicely confirms the presence of the expected 9-mer product.”

7. Line 210: It is unclear what “they” refers to.

We agree that this sentence needed clarification. It has been updated as such: “The ability to visualize a faint signal in the absence of any primer prompted us to analyze whether the products were mono-phosphate as previously proposed⁶.”

8. Supp Fig. 7 would benefit from the same changes as Fig. 2/Supp. Fig. 4.

We have added on **Supplementary Fig. 10** an overlay of the two structures with a common portion aligned with arrows and distances to show the direction and magnitude.

9. Lines 268-274: Much of this seems unnecessary and could be moved to supplemental text.

We have deleted lines 268-271. Lines 272 to 274 are necessary as the identification of Y1564 as the residue of the lid that separates the template and the product is an important feature of the elongation state.

10. Line 318: 5' → 5'vRNA

This has been corrected in “5'vRNA₁₋₂₅” as requested in main point 6.

11. Line 349: infirming → disproving?

Thank you. This was a wrong translation. It has been updated into “contradicting”.

12. Figure 1: text is too small (in a.), some colors as well

Text of **Fig. 1a** has been increased in size. Colors have been checked and we did not see a problem (this point is not clear: colors cannot be too small).

13. Figure 2: a,b) Why does the size of the EMSA products shift down in the presence of poly(A)? This seems unexpected.

We interpret that the addition of the poly(A) (of long molecular weight and heavily negatively charged) necessarily causes a faster migration of the HTNV-L/vRNA/poly(A) complexes in an electrophoresis agarose gel. What our experiments suggest is that HTNV-L is able to bind poly(A) RNA non-specifically in the presence or absence of a specific RNA molecule bound to HTNV-L (e.g. the poly(A) molecules could be binding inside the inner chamber still allowing 5'vRNA binding in its specific binding site). But what is important in this experiment is that, even if the specific RNA and the poly(A) binding can occur simultaneously, the poly(A) displaces unspecific RNA binding, thus demonstrating binding specificity.

b) I'm very confused by the ss and ds labels. How is that validated?

We have removed the ss and ds labels in the figure and now refer to “RNA/protein complexes” and “RNA free”.

14. Figure 3: b) MW25 ladder is very difficult to interpret. Consider redoing these experiments with a different ladder that is clearer to interpret.

We have redone the gel with a less intense marker as we agree it was too intense to be interpreted easily. This marker is the best we can have as it has the correct sequence and therefore migrate as the products.

As Reviewer 2 was requesting a different ladder we have also included the commercial decade molecular weight marker. However, please be aware the sequence of the RNA in the decade marker is not the same as in the product and so the marker does not migrate exactly as the product. We are open to discussion with Nature Communications Editorial team to know if we should put the decade marker on the final **Fig. 2b** or not.

15. Figures 4&5: Yellow text is very difficult to read, add a black border around the text. Same with most of the base labels in 4c, 5d.

Nature communications does not allow to have black border around the text at the editing stage. We have modified the color of the yellow, making it darker, for better visualization.

Reviewer #3 (Remarks to the Author):

The last couple of years, the RNA polymerase polymerases of various negative sense RNA viruses have been extensively studied using crystallography and cryo-EM. These studies revealed the various steps of the transcription or replication cycle, and identified many shared principles. Here, Trouillette et al present the near-complete replication cycle, split over 5 cryo-EM structures, of a hantavirus RNA polymerase. The authors employ biochemical experiments and previously explored promoter mutations to study the initiation and elongation activities of the hantavirus RNA polymerase in vitro. Overall, the experiments are well-designed, the structures insightful and carefully annotated, and the manuscript, with the exception of a handful of grammatical errors, very clearly written. While other studies provide insight into many aspects of the nucleotide addition cycle of negative sense RNA virus RNA polymerases and in particular the transcription cycle, this study is a tour-the-force on the hantavirus RNA polymerase replication cycle. The data reported here will be useful for our fundamental understanding of negative sense virus replication, the appreciation of unique characteristics of different viral replication complexes (some of which the authors identify for the hantavirus RNA polymerase), and the development of broad-spectrum as well as virus-specific antivirals going forward. I don't have any substantive concerns about this manuscript.

We thank Reviewer 3 for the very positive comments about the data, their interpretation, the clarity of the manuscript. The article has now been corrected by a native English speaker to try removing the handful of grammatical errors that were present.

Reviewer #1 (Remarks to the Author):

I want to thank the authors for addressing the initial comments. Following the revision to the article, I do not have more questions now.

Reviewer #2 (Remarks to the Author):

The authors have addressed all our concerns and the manuscript is ready for publication.